# Simulating secondary organic aerosol from anthropogenic and biogenic precursors: comparison to outdoor chamber experiments, effect of oligomerization on SOA formation and reactive uptake of aldehydes

Florian Couvidat[1], Marta G. Vivanco[2], and Bertrand Bessagnet[1]

[1]Institut National de l'Environnement Industriel et des Risques, Verneuil-en-Halatte, France
[2]Centro de Investigaciones Energéticas, Medioambientales y Tecnológicas (CIEMAT), Departamento de Medio Ambiente, Av. Complutense 40, 28040-Madrid, Spain

*Correspondence to:* Florian Couvidat
(Florian.Couvidat@ineris.fr)

**Abstract.**

New parameterizations for the formation of organic aerosols were developed. These parameterizations cover SOA formation from biogenic and anthropogenic precursors, $NO_x$ dependency, oligomerization and the reactive uptake of pinonaldehyde. These parameterizations were implemented in a box model where the condensation/evaporation of semi-volatile organic compounds was simulated by the Secondary Organic Aerosol Processor (SOAP) model to take into account the dynamic evolution of concentrations.

The parameterizations were tested against several experiments carried out in previous studies in the EUPHORE outdoor chamber. Two datasets of experiments were used: the anthropogenic experiments (where SOA is formed mainly from a mixture of toluene, 1,3,5-trimethylbenzene and o-xylene) and the biogenic experiments (where SOA is formed mainly from $\alpha$-pinene and limonene).

When assuming no wall deposition of organic vapors, satisfactory results (bias lower than 20%) were obtained for the biogenic experiments and for most of the anthropogenic experiments. However, a decrease of SOA concentrations (up to 30%) was found when taking into account wall deposition of organic vapors (with the parameters of Zhang et al. (2014)). The anthropogenic experiments seem to indicate a complex $NO_x$ dependency that could not be reproduced by the model. Oligomerization was found to have a strong effect on SOA composition (oligomers were estimated to account for up to 78% of the SOA mass) and could therefore have a strong effect on the formation of SOA. The uptake of pinonaldehyde (which is a high volatility SVOC) onto acidic aerosol was found to be too slow to be significant under atmospheric conditions (no significant amount of SOA formed after 3 days of evolution), indicating that the parameterization of Pun and Seigneur (2007) used in some air quality models may lead to an overestimation of SOA concentrations. The uptake of aldehydes could nevertheless be an important SOA formation pathway for less volatile or more reactive aldehydes than pinonaldehyde.

Regarding viscosity, a low effect of viscosity on SOA concentrations was estimated by the model, although a decrease

of SVOC evaporation was found when taking it into account, as well as a lower sensitivity of concentrations to changes of temperature during the experiments.

# 1 Introduction

Because of the effect of fine particles on human health (WHO, 2003) and ecosystems (Kanakidou et al., 2005), the use of models have become a common practice to evaluate impacts and mitigation strategies. Particulate organic matter (OM) represents a large fraction of the total fine particulate mass, typically between 20 and 60% (Kanakidou et al., 2005; Yu et al., 2007; Zhang et al., 2007), with the secondary fraction (secondary organic aerosols, SOA) representing most of it (90% according to the best estimate of Kanakidou et al. (2005)). Therefore, efforts have to be made to represent OM as accurately as possible in models.

Numerous models have been developed to simulate OM in 3D air quality models (Schell et al., 2001; Donahue et al., 2006, 2011; Pun et al., 2002; Couvidat et al., 2012; Griffin et al., 2003; Jathar et al., 2015; Tulet et al., 2006; Carlton et al., 2010; Menut et al., 2013). Most of these models use simple parameterizations based on SOA yields estimated from smog chamber experiments conducted under specific conditions, which can be different from atmospheric conditions (low humidity, specific $NO_x$ conditions). Nevertheless, SOA yields from chamber experiments have traditionally been estimated considering particle wall-losses, but not gas wall-losses. Several studies (Zhang et al., 2014, 2015; Bian et al., 2015; Cappa et al., 2016) indicate that this fact could have led to a strong underestimation of SOA yields. On the other side, none of these models take into account the whole complexity of the processes involved in organic aerosol formation (non-ideality, multi-phase partitioning, viscosity of the aerosol, phase separation, aging, oligomerization and organosulfate formation, effects of $NO_x$ concentrations, etc...), that can highly affect the level of SOA predicted (e.g. Ng et al. (2007a, b); Pun and Seigneur (2007); Couvidat and Seigneur (2011); Hall IV and Johnston (2011)). Therefore, the development of parameterizations addressing these aspects can provide insights about the SOA formation processes, and improve current model estimates.

Oligomer formation has been addressed in some modeling studies. This process may be important for organic aerosol as it can transform semi-volatile and volatile organic compounds into less volatile compounds. Trump and Donahue (2014) studied the effect of oligomerization on the dynamic of organic aerosol formation. Pun and Seigneur (2007) developed a parameterization for the oligomerization of aldehydes by increasing their partitioning toward the particle. This parameterization (treating oligomerization with an equilibrium constant) have been used in Couvidat et al. (2012) to increase SOA formation from pinonaldehyde. High aerosol concentrations were simulated with this parameterization. However, Liggio and Li (2006b) showed that the uptake of pinonaldehyde onto acidic aerosol is a slow process and that this process could be due to oligomerization but also to organosulfate formation. Carlton et al. (2010) used a simple first-order rate constant of oligomerization for all organic compounds based on the results of Kalberer et al. (2004). Jathar et al. (2016) used the parameterization of Carlton et al. (2010) and showed that, depending on the mechanism, it could have a strong effect or not on SOA formation. In all the cases, the authors found a strong effect on composition (or volatility distribution). Lemaire et al. (2015) compared these different approaches for oligomerization and emphasized the need to simulate properly oligomerization in air quality models.

Several approaches have been used to represent SOA formation in 3D air quality models. Most of these approaches have

in common that they are based on results from smog chamber experiments but differ in the methodology for lumping organic compounds and the treatment of the processes involved in SOA formation. Among these approaches, the Volatility Basis Set (VBS) approach (Donahue et al., 2006; Robinson et al., 2007) has been widely used (Lane et al., 2008; Shrivastava et al., 2008; Murphy and Pandis, 2009; Shrivastava et al., 2011; Han et al., 2016). It uses a basis set where Semi-Volatile Organic Compounds (SVOC) are distributed on an equally spaced logarithmic scale for volatility (volatility bins). Aging was treated in the VBS by assuming a shift of volatility per aging step (gas-phase oxidation creates a change of volatility bins). A 1.5D VBS (Koo et al., 2014) and a 2D VBS (Donahue et al., 2011) were also developed to take into account changes in the oxidation degree.

Another approach used in 3D air quality models is the molecular surrogate approach (Pun et al., 2002, 2006; Griffin et al., 2003; Couvidat et al., 2012). In this approach, surrogate compounds (that gather a large number of SVOC with similar thermodynamic properties) are associated with molecular structures to extrapolate SOA formation from smog chambers to the atmosphere. In the molecular surrogate approach, several processes, which are often not taken into account in the other approaches can be readily estimated (e.g., absorption into an aqueous phase, hygroscopicity, non-ideality) and can be treated explicitly in the model. Based on this methodology, the Secondary Organic Aerosol Processor (SOAP) (Couvidat and Sartelet, 2015) was developed to simulate the gas/particle partitioning of SVOC by taking into account non-ideality (by considering the interactions between organic and inorganic compounds), multi-phase partitioning, phase separation, hygroscopicity and the viscous state of organic aerosols. It was recently implemented in the CHIMERE air quality model (Couvidat et al., 2018) to simulate the gas/particle partitioning of SVOC formed with the $H^2O$ mechanism (Couvidat et al., 2012). However, in $H^2O$, a simplified aging mechanism was only applied for the first step of oxidation of anthropogenic primary SVOC due to lack of information and SVOC were supposed to be only hydrophilic (condense only the aqueous phase of particles) or only hydrophobic (condense only the organic phase of particles). Moreover, for anthropogenic SOA, comparison to measurements carried out inside Euphore has shown that for anthropogenic SOA the $H^2O$ mechanism may overestimate SOA in the conditions of the experiments.

In order to create a SOA mechanism that could be implemented in 3D air quality models and that considers the main phenomena involved in SOA formation (non-ideality, multi-phase partitioning, viscosity of the aerosol, phase separation, aging, oligomerization, effects of $NO_x$ concentrations), the aim of this study is to improve and update the $H^2O$ mechanism for the formation of SOA from the oxidation of monoterpenes and aromatics by reconsidering some of the assumptions made in the development of this mechanism (no aging, compounds only hydrophilic or only hydrophobic, use of the parameterization of Pun and Seigneur (2007)) and have a more realistic representation of SOA formation. The update of the mechanism was done based on SOA yields available in the literature with the methodology of Odum et al. (1996); an aging mechanism was added; SVOC surrogates are assumed to be able to condense on both the aqueous phase and the organic phase. In order to evaluate this SOA mechanism, it was tested against some experiments carried out in the EUPHORE outdoor chamber under conditions different from the conditions under which it was developed (mixture of different compounds, non-controlled temperature) to ensure the good performances of the model.

Two new parameterizations on particle-phase reactions were also developed covering oligomerization and the uptake of

pinonaldehyde. For oligomerization, instead of representing this process by a first-order complete reaction as done by Carlton et al. (2010), it was chosen to reproduce this phenomena (as an effort to provide a more realistic representation) with a second-order reversible reaction unfavored by humid conditions by analogy to oligomerization reactions expected to occur in the particle like esterification, hemiacetalization, aldolization, peroxyhemiacetalization (Roldin et al., 2014). For the uptake of pinonaldehyde, instead of assuming equilibrium (as done by Pun and Seigneur (2007)), a parameterization for the dynamic of the uptake, which was shown to limit strongly the uptake of pinonaldehyde (Liggio and Li, 2006b) as function of the particle acidity, was taken into account.

In addition, the effect of particle phase reactions was investigated. Oligomerization was simulated with the second-order reversible reaction to evaluate the impact of this process and how it may impact SOA formation. For the uptake of pinonaldehyde onto acidic aerosols, the results obtained with the equilibrium based parameterization of Pun and Seigneur (2007) were also compared to the results given by the dynamic parameterization developed in this study. SOA concentrations simulated with these parameterizations were compared to measured concentrations from experiments in the presence of $SO_2$ (that leads to the formation of acidic aerosols). Finally, the effects of low diffusion inside the particle due to high viscosity and of wall losses of organic vapors were also studied.

## 2 Method

The new parameterizations described hereafter were developed and tested against experiments that were carried out in previous studies inside the outdoor chamber EUPHORE in Valencia.

### 2.1 Experimental datasets

EUPHORE is a 204 m$^3$ hemispherical Teflon outdoor chamber located in Valencia, (CEAM), Spain. The chamber is surrounded by a retractable steel housing, which can be opened or closed to control the time of exposure to sunlight. The housing also serves to protect the chamber from rain and strong winds. The floor of the reactor consists of aluminum panels covered with Teflon, with a cooling system designed to compensate the heating of the chamber caused by solar irradiation. Two high-powered mixing fans (with an air flow of 4000 m$^3$/h), located on the floor of the chamber, are used to ensure the homogeneous mixing of reactants and products. The EUPHORE facility is described in detail in Volkamer et al. (2001).

The experiments used for the comparison between the model and experiments have been published in previous studies (Vivanco et al., 2011, 2013). The experiments used in this study were initially planned to increase the experimental information regarding SOA formation and to be mainly used in model performance evaluation processes. As most of the studies at that time were focused on the oxidation of individual organic gases or simple mixtures of them, those experiments constitute a contribution to the experimental database on SOA formation, by considering the oxidation of different mixtures of organic precursors. Although the experimental dataset was designed for model evaluation purposes, it was not designed to evaluate the model developed in this study. Vivanco et al. (2016) and Santiago et al. (2012b) used these experiments to evaluate simple parameterizations existing in several air quality models. The experiments may not cover the full range of values covered by the

model for some parameters, especially for relative humidity (RH) which was lower than 40% for all experiments. Experimental conditions are described in Tables 1 and 2.

OH was formed by HONO photolysis. Ozone was formed throughout the experiment and often reached concentrations around 200 ppb. However, at the start of the experiments where concentrations of ozone are lower, monoterpenes will react mainly with OH radicals. As almost all the monoterpenes reacted during the first hour of the reaction, monoterpenes can be considered to react both with OH and $O_3$. Calculations carried with RACM2 showed that 10 to 70% of the monoterpenes should have reacted with $O_3$. For experiments B4 and B5 with very low relative humidities (around 10%), most of the monoterpenes reacted with OH (around 90%). OH photo-oxidation time (determined for a concentration of $1.5 \times 10^6$ molecules cm$^{-3}$) was determined to range from a a few hours to more than a day.

Two datasets of experiments are used: (i) the anthropogenic experiments where SOA is formed from the oxidation of toluene (TOL), 1,3,5-trimethylbenzene (TMB) and to a lesser extent o-xylene and octane and (ii) the biogenic experiments where SOA is formed from $\alpha$-pinene (API) and limonene (LIM) and to a lesser extent isoprene. Experimental conditions for the anthropogenic and biogenic experiments are described respectively in Tables 1 and 2. For the biogenic dataset, two experiments were carried out in presence of $SO_2$ to evaluate the parameterization of Pun and Seigneur (2007). Due to its low SOA yield, SOA from isoprene oxidation should not represent a significant amount of total SOA.

Although octane is a precursor of SOA and is present in the anthropogenic experiments, SOA formation from octane oxidation was not taken into account in this study as Vivanco et al. (2016) showed with an experiment in the same chamber that octane leads to an insignificant amount of SOA. Due to its low yield (0.5% according to Lim and Ziemann (2005)), octane SOA should not represent more than a few percent of total SOA.

PM volume concentrations were measured with a Scanning Mobility Particle Sizer Spectrometer (SMPS). Data are not corrected for wall losses as deposition of particles is taken into account in the model.

## 2.2 Model development

Secondary organic aerosols inside the chamber were simulated by coupling the gas-phase mechanism RACM2 (Goliff et al., 2013) with the SOAP model (Couvidat and Sartelet, 2015) to compute the dynamic formation of SOA. RACM2 was used because it has been shown to perform well for oxidant formation (Kim et al., 2009). To represent the chemical evolution of SVOC, RACM2 was modified to take into account the formation of the surrogate species according to the mechanisms described hereafter. The ROS2 algorithm (Verwer et al., 1999) was used to solve the chemical kinetic equations.

SOAP is a model designed to be modular with user options depending on the computation time. SOAP uses the molecular surrogate approach to estimate several properties and parameters (hygroscopicity, absorption into the aqueous phase of particles, activity coefficients and phase separation) and to evaluate the partitioning of organic compounds between one or several organic phases (the number of organic phases is determined by Gibbs energy minimization) and the inorganic phase. It accounts for the influence of interactions between organic and inorganic compounds by using the AIOMFAC algorithm (Zuend et al., 2008, 2011; Zuend and Seinfeld, 2012). Secondary inorganic aerosol formation was added to the SOAP model by using

the equilibrium parameters of ISORROPIA v2.1 (Fountoukis and Nenes, 2007).

Currently, SOAP assumes that inorganic aerosols are metastable liquids and therefore does not take into account efflorescence or deliquescence processes. This assumption could be wrong in presence of ammonia due to the low humidity inside the chamber, as ammonium sulfate would probably be solid as the humidity in the chamber (below 40%) is always far below the deliquescence relative humidity (80% at 298K). However, no ammonia was present in these experiments and $SO_2$ was only introduced for two experiments. For those two experiments, $SO_2$ oxidation will lead to sulfuric acid formation which remains liquid over the full RH range (Seinfeld and Pandis, 1998).

Several experimental data were used to constrain the model. Temperature and relative humidity measurements inside the chamber were used as inputs for the model. SMPS measurements were used to compute the mean diameter of particles. As modeling properly nucleation and coagulation of particles would be needed to simulate adequately the size distribution of particles, particles were gathered inside a single diameter bin. The mean diameter of particles was constrained to provide the model with a realistic estimation of the diameter without modeling the nucleation in SOAP. The diameter of particle is used to compute the kinetic rate of condensation/evaporation/diffusion and the Kelvin effect by the dynamic approach of SOAP.

The model results were compared to non-corrected (according to wall losses) results. Wall deposition of particles was simulated via first order loss rate parameter constrained to reproduce the loss of particles during the last hours of the experiment when the chamber was enclosed by the retractable steel housing.

Wall losses of vapors were not taken into account due to the lack of information on deposition onto EUPHORE walls. However, in the last section, the effect of vapor wall losses on SOA formation in the chamber was investigated by taking into account vapor wall losses according to parameters estimated by Zhang et al. (2014).

### 2.2.1 SOA Mechanism

A new mechanism was developed for SOA formation from toluene (TOL), o-xylene (XYL) and trimethylbenzene (TMB). Parameters were fitted on data issued from several studies under low and high $NO_x$ conditions: Ng et al. (2007b) for Toluene SOA, Cocker III et al. (2001) for TMB SOA and the results from both studies for for Xylene SOA. No information was found on SOA formation from TMB oxidation under low-$NO_x$ conditions, therefore, the stoichiometric coefficient from the low-$NO_x$ conditions for xylene was used. Similarly to Vivanco et al. (2016), the low-$NO_x$ condition yields are used only if radicals formed from the oxidation of the precursors react at least twice with the $HO_2$ radical, to prevent high formation of low-$NO_x$ SOA under intermediate $NO_x$ conditions. Molecular structures (used by SOAP to estimate several properties and to compute activity coefficients) were selected based on the results of Im et al. (2014) by selecting the compound with a similar volatility, formed in the largest quantity and ensuring the best reproduction of the O/C and H/C ratios. Properties of surrogate species and reactions are shown in Table 3. Reactions leading to SOA formation are shown in Table 4 for TOL SOA and in Table 5 for XYL and TMB SOA.

For monoterpenes, the mechanism of Pun et al. (2006) was updated using results from more recent studies. For SOA formation from $\alpha$-pinene, $\beta$-pinene and limonene ozonolysis, the mechanism is based on the parameterizations described in Lee et al. (2011) for high and low $NO_x$ conditions. The API + OH reaction is based on the results of Svendby et al. (2008) whereas

for the LIM + OH reaction, the yield of BiA2D was optimized to give the best results. Reactions and properties of surrogate species are shown in Tables 6 and 3. The terpene + $NO_3$ yields are based on Fry et al. (2014).

In the surrogate SOA approach, results of the model may depend on the choice of the surrogate structure to represent SOA formation. Although different VOCs form different compounds (or the same compounds in different quantities), this approach lumps together species that have similar properties (like saturation vapor pressure) and that are expected to have similar chemical structure. For example, first-generation aldehyde products from terpene oxidation were represented by the surrogate species BiA0D. Moreover, the compounds were selected (based on the best information available on SOA products) to reproduce the mean properties of SOA and of the O/C and H/C ratios. Kim et al. (2014) showed that the mean O/C and H/C ratios are around 0.34-0.36 and 1.4-1.5 for SOA from $\alpha$-pinene and limonene respectively. The selected surrogates give a O/C between 0.3 and 0.44 and a H/C between 1.55 and 1.6. For TOL SOA, the chosen surrogates give a O/C ratio (between 0.71 and 0.8) and H/C ratio (between 1.14 and 1.6) similar to the reported values by Schilling (2015) (between 0.7 and 0.8 for O/C and between 1.2 and 1.6 for H/C). For TMB SOA, Sato et al. (2012) indicate a O/C ratio around 0.4-0.6 and a H/C around 1.5-1.7. The chosen surrogates reproduce the O/C ratio (between 0.38 and 0.55) but seem to underestimate the high H/C ratio (between 1.13 and 1.44 due to the low H/C of the AnIP2 compound). However, for AnIP2, a better molecular structure having a similar O/C ratio and saturation vapor pressure could not be found.

The mechanism of Couvidat and Seigneur (2011) was used to simulate SOA formation from isoprene.

### 2.2.2 Oligomerization

Carlton et al. (2010) developed a parameterization for oligomerization based on a simple first-order rate constant of oligomerization. Based on the results of Kalberer et al. (2004) indicating 50% of polymers after 20h in TMB SOA, Carlton et al. (2010) determined a kinetic constant of $9.6 \times 10^{-6}$ $s^{-1}$. However, this number was obtained with a laser desorption ionization–mass spectrometry (LDI-MS) by taking all compounds with m/z higher than 400 (with m the ion mass and z the ion charge) and may not take into account small oligomers (dimers or trimers that may be formed more rapidly). Kalberer et al. (2004) and Kalberer et al. (2006) also studied the oligomer fraction based on a volatility tandem differential mobility analyzer (VTDMA) giving the remaining volume fraction of particles at different temperatures. They found that for TMB SOA, the remaining fraction at 100°C after 5h ranges from 50% to 62% and is composed of small oligomers or very low volatility organic compounds. The remaining fraction at 100°C can even reach 80% to 90% after 25h. Based on these results and assuming that the remaining fraction is mainly composed of small oligomers and that oligomerization is irreversible, the first-order constant of oligomerization should be around $3.85 \times 10^{-5}$ $s^{-1}$ to take into account small oligomer formation.

However, a first order complete reaction may not be appropriate to represent oligomerization. Indeed, Roldin et al. (2014) showed that oligomerization should involve second order reversible reactions like esterification, hemiacetalization, aldolization, peroxyhemiacetalization. The equilibrium of these reactions are unfavored by humid conditions and the reaction is catalyzed under acidic conditions. Indeed, oligomer formation by esterification was reported in the case of isoprene SOA (Surratt et al., 2006).

In this study, oligomerization is represented by a reversible process which is mainly due to mechanisms like esterification,

unfavored by humid conditions. It is represented by either a simple "reduced" reaction (or "bulk oligomerization") or an extended representation. For the extended representation, each of the oligomerization step (each combination up to the formation of tetramer) is represented:

$$A + A \underset{k_{reverse,extended}}{\overset{k_{oligo,extended}}{\rightleftarrows}} A_{dimer}$$

$$A + A_{dimer} \underset{k_{reverse,extended}}{\overset{k_{oligo,extended}}{\rightleftarrows}} A_{trimer}$$

$$A + A_{trimer} \underset{k_{reverse,extended}}{\overset{k_{oligo,extended}}{\rightleftarrows}} A_{tretramer}$$

$$A_{dimer} + A_{dimer} \underset{k_{reverse,extended}}{\overset{k_{oligo,extended}}{\rightleftarrows}} A_{tretramer} \tag{1}$$

with A a monomer compound, $A_{dimer}$ the dimer, $A_{trimer}$ the trimer of compound A and $A_{tretramer}$ the tetramer of compound A, $k_{oligo,extended}$ the forward kinetic of oligomerization and $k_{reverse,extended}$ the reverse kinetic of oligomerization. In the case of a mixture of several compound, each combination and each product would need to be represented. The same value of $k_{oligo,extended}$ and $k_{reverse,extended}$ are used for each steps. If more data were available, the parameterization could be improved to take into account different kinetic rate parameters of oligomerization per combination. The forward kinetic of oligomerization can be linked to the reverse kinetic of oligomerization via the oligomerization equilibrium constant:

$$\frac{k_{oligo,extended}}{k_{reverse,extended}} = \frac{(K_{oligo}^{eq})}{(a_{H_2O})} \tag{2}$$

For the "reduced" (or "bulk oligomerization") parameterization, all the steps are described by a single reaction:

$$A \underset{k_{reverse}}{\overset{k_{oligo}}{\rightleftarrows}} \frac{1}{m_{oligo}} A_{oligo} \tag{3}$$

with A a monomer compound, $A_{oligo}$ the monomer blocks of compound A inside oligomers, $m_{oligo}$ the number of monomer blocks inside oligomers, $k_{oligo}$ the kinetic rate parameter of oligomerization and $k_{reverse}$ the kinetic rate parameter of the reverse reaction. In this parameterization, oligomers are represented by a single model species. A represent the monomer and $A_{oligo}$ any monomer blocks present inside the oligomers. It can be use to represent oligomerization in a mixture of compounds (for example a mixture of compounds A and B). If a monomer block from a compound A reacts with another monomer block (the same compound A or another compound B), the compound will be converted from monomer A to oligomer $A_{oligo}$ (independently of the reaction with A or B). The parameterization currently does not take into account different kinetic rate parameters between each combination of compounds due to lack of data. It assumes that a compound A reacting with itself and a compound B reacting with itself will have the same kinetic parameter and will have the same value when compounds A and B react together. If more data were available, the parameterization could be improved to take into account different kinetic rate parameters of oligomerization per combination.

In this study, the net flux of oligomerization $J_{oligo}$ is computed using activities. Activity is often seen as the "apparent concentration" of a compound in thermodynamics. It is linked to the chemical potential (molar Gibbs free energy of a particular

component) by the following equation:

$$a_i = exp(\frac{\mu_i - \mu_i^0}{RT})$$ (4)

with $a_i$ the activity of compound i, $\mu_i$ is the chemical potential of compound i and $\mu_i^0$ the chemical potential under standard conditions, R is the gas constant, T is thermodynamic temperature.

Activities (calculated here on the mole fraction basis) are used instead of concentrations for two main reasons. First, chemical rates are more consistent with thermodynamic equilibrium by computing rates using activities. For example, in the case of a simple one product (A) giving one product (B) equilibrium reaction, if chemical reactions are written using concentrations, the net flux of reaction J would be computed with the following equation:

$$J = k_1 C_A - k_{-1} C_B$$ (5)

with $k_1$ the forward kinetic parameter, $k_{-1}$ the reverse kinetic parameter, $C_A$ the concentration of compound A and $C_B$ the concentration of compound B. At equilibrium, J would be equal to zero and the equilibrium constant would then correspond to the ratio of concentrations instead of a ratio of activities. This paradox can be lifted by using activities instead of concentrations. Second, some studies (Madon and Iglesia, 2000; Rahimpour, 2004) expressed the need to compute chemical rates using activities and showed that better results are obtained for non-ideal systems.

The net flux of oligomerization $J_{oligo}$ for the reduced parameterization (Eq. 3) is therefore computed with the following equations:

$$J_{oligo} = -\frac{dX_{a,monomer}}{dt} = k_{oligo} a_{a,monomer} - k_{reverse} a_{a,oligomer}$$ (6)

with $X_{a,monomer}$ the molar fraction of compound a, $a_{a,monomer}$ the activity on a molar fraction basis of compound a and $a_{a,oligomer}$ the activity on a molar fraction basis of the oligomer formed from compound a. Activities are computed with the 20    AIOMFAC model (Zuend et al., 2008, 2011; Zuend and Seinfeld, 2012; Ganbavale et al., 2015).

The kinetic rate of oligomerization $k_{oligo}$ is computed as follows:

$$k_{oligo} = k_{oligo}^{max} a_{monomer}$$ (7)

with $k_{oligo}^{max}$ the maximum kinetic rate parameter for oligomerization, and $a_{monomer}$ the sum of monomer activities.

To calculate the reverse kinetic rate parameter, the computation is based on the equilibrium oligomerization constant $K_{oligo}^{eq}$. 25    The equilibrium constant for oligomerization (due to esterification or a similar oligomerization mechanism) is computed with:

$$(K_{oligo}^{eq})^{m_{oligo}-1} = \frac{a_{a,oligomer}(a_{H_2O})^{m_{oligo}-1}}{a_{a,monomer}(a_{monomer})^{m_{oligo}-1}}$$ (8)

with $a_{H_2O}$ the activity of water on a molar basis.

At equilibrium, the rate of oligomerization is zero. Therefore,

30    $$\frac{k_{oligo}^{max}}{k_{reverse}} = \frac{a_{a,oligomer}/m_{oligo}}{a_{monomer} a_{a,monomer}} = \frac{(K_{oligo}^{eq})^{m_{oligo}-1}(a_{monomer})^{m_{oligo}-2}}{(a_{H_2O})^{m_{oligo}-1}}$$ (9)

To represent oligomerization, $k_{oligo}^{max}$, $m_{oligo}$ and $K_{oligo}^{eq}$ were estimated based on the results of Kalberer et al. (2006). In their study, the molar masses of heavy molecules in SOA from several precursors were measured using matrix-assisted laser desorption/ionization mass spectrometry (MALDI-MS). To represent oligomerization, we represented explicitly with the extended parameterization (Eq. 1) the formation of oligomers up to tetramers from a single monomer to simulate the evolution of mass averaged molar mass of oligomers (which cannot be simulated with our reduced parameterization (Eq. 3) for oligomerization because the molar mass of oligomers does not vary with this parameterization). The equilibrium constant of oligomerization $K_{oligo}^{eq}$ and the oligomerization constant $k_{oligo,extended}$ of the extended parameterization (Eq. 1 and Eq. 2) were fitted so that the evolution as a function of time of the weight average molar mass of oligomers during the first hours of the experiment is reproduced by the model as shown by Fig. 1. A molar mass of the TMB SOA monomer of 155 g/mol was chosen so that the molar mass of TMB SOA is between 140 and 170 g/mol based on Im et al. (2014). $m_{oligo}$ and $k_{oligo}^{max}$ are chosen so that the molar averaged mean molar mass and the total mass of oligomers (with oligomers and monomers) of the reduced (Eq. 3) and the extended (Eq. 1) parameterizations were similar. The reduced parameterization values (Eq. 3) found for $k_{oligo}^{max}$, $m_{oligo}$ and $K_{oligo}^{eq}$ were respectively $2.2 \times 10^{-4}$ s$^{-1}$, 3.35 and 2.94 for TMB SOA. $k_{oligo}^{max}$ is in the same order of magnitude as the kinetic rate parameter reported for the reaction acetaldehyde and methanol by Roldin et al. (2014): $4.9 \times 10^6$ a[H$^+$] M h$^{-1}$ (a[H$^+$] being the activity of H$^+$), which should be around $6 \times 10^{-4}$ s$^{-1}$ on an activity basis for a pH of 4.6 (order of magnitude of pKa for carboxylic acids). Eq. 3 could be also used to take into acoount the oligomerization inside an aqueous acidic phase if the effect of acidity is taken into account. Based on a pH of 4.6, a kinetic rate of 8.76 a[H$^+$] (instead of $k_{oligo}^{max}$ in Eq. 3) should be used to take into account the effect of acidity on oligomerization.

This reduced parameterization (Eq. 3) only takes into account the formation of short oligomers (oligomers of 2 to 4 monomers blocks that can be formed quickly during the first hours) but should give a good insight on the impact of oligomerization on SOA formation. Formation of short oligomers can impact SOA formation by transforming monomers and increasing the condensation of semi-volatile compounds onto the particle. However, big oligomers (more than 4 blocks of monomers) formation should affect less SOA formation as it should mainly lead to the transformation of oligomers into bigger oligomers (oligomers with higher molar masses). Big oligomers could affect indirectly the partitioning of monomers by increasing the mean molar mass of the organic phase; the partitioning constant of monomers being inversely proportional to the mean molar mass.

This parameterization also gives good results for the isoprene SOA oligomerization using a molar mass of 120 g/mol. This molar mass corresponds to the molar mass of methyl glyceric acid, which was shown to undergo oligomerization by esterification. For $\alpha$-pinene, Kalberer et al. (2006) did not find any significant temporal evolution of oligomer molar masses, which is consistent with the formation of dimers that cannot react further and therefore parameters for oligomerization of $\alpha$-pinene SOA cannot be evaluated.

The results for SOA oligomerization from TMB and isoprene for the extended (Eq. 1) and reduced parameterizations (Eq. 3) are shown in Fig. 1. For the formation of dimers from $\alpha$-pinene, we used the same parameters except that $m_{oligo}$ is set to 2 to limit the formation of oligomers to dimers. DePalma et al. (2013) confirmed that particle-phase dimer formation is possible. However, Kristensen et al. (2014) studied the formation of 4 dimers and determined that those 4 dimers are not formed from

particle-phase reaction but through gas-phase reactions of the stabilized Criegee Intermediate formed from the ozonolysis of $\alpha$-pinene. It could then be possible that in the case of $\alpha$-pinene not all the oligomers are formed via particle-phase reactions.

In this study, the second order "reduced" parameterization (Eq. 3) was used for simulations.

### 2.2.3 Uptake of pinonaldehyde onto acidic aerosols

Several studies (Liggio and Li, 2006a, b) reported an uptake of pinonaldehyde onto acidic aerosols higher than what could be predicted by assuming equilibrium between the gas and particle phases and no chemical reaction inside the particles. This phenomenon is attributed to oligomer and/or organosulfate formation. Gao et al. (2004b) reported a similar phenomenon for various aldehydes. To represent this phenomenon, Pun and Seigneur (2007) developed a parameterization by computing an effective Henry's law constant $H_{eff}$ for aldehyde compounds:

$$H_{eff} = H \left( 1 + 0.1 \left( \frac{a(H^+)}{10^{-6}} \right)^{1.91} \right) \tag{10}$$

where H is the monomer Henry's law constant of the compound, and a($H^+$) is the activity of protons in the aqueous phase. As fine particles are generally very acidic (Ludwig and Klemm, 1990; Keene et al., 2004), uptake of pinonaldehyde will in fact appear to be an irreversible process, even though the parameterization is formulated as a reversible process. Using this parameterization for pinonaldehyde in 3D air quality models leads to very high concentrations of SOA from monoterpenes
(Couvidat et al., 2012).

However, this parameterization does not take into account the uptake rate of aldehydes. Liggio and Li (2006b) measured the uptake rate coefficients for various acidities of the aerosols. The authors showed that the uptake of pinonaldehyde onto aerosols can only be significant for very high acidities (which can be reached with low ammonia concentration and at low humidities). Based on these results, a parameterization for the chemical evolution of pinonaldehyde has been developed in this study:

$$BiA0D_{part} \xrightarrow{k_{trans}} \text{Non volatile products} \tag{11}$$

with BiA0D the surrogate species used by Couvidat et al. (2012) to represent pinonaldehyde, $BiA0D_{part}$ the amount of pinon-aldehyde inside the particle and $k_{trans}$ (in $s^{-1}$), the kinetic rate of BiA0D transformation inside the particle into a product assumed non-volatile. Liggio and Li (2006b) did not evaluate a kinetic rate of pinonaldehyde transformation inside the particle but measured a kinetic rate of uptake. However, the kinetic parameter of uptake $k_{uptake}$ can be linked to the kinetic parameter of transformation $k_{trans}$ by assuming equilibrium between the gas and particle phases:

$$k_{uptake} = k_{trans} K_{aq} AQ \tag{12}$$

with $K_{aq}$ the partition coefficient of pinonaldehyde between the gas phase and the particle and AQ the particle mass.

As in Liggio and Li (2006b), the pH of particles and activities of compounds were computed with AIOMFAC (Zuend et al., 2008, 2011; Zuend and Seinfeld, 2012; Ganbavale et al., 2015) depending on the conditions (humidity, temperature,
concentrations, etc...) of the experiments of Liggio and Li (2006b). The estimated flux of transformation based on the results

of Liggio and Li (2006b) was plotted against several variables to find the variables exhibiting the best correlations. $m_{H^+}$ the molality of ion $H^+$ and $a^{(m)}_{HSO_4^-}$ the activity as a molality basis of ion $HSO_4^-$ were found to be the best variables with high correlation coefficients (0.98 for $m_{H^+}$ and 0.92 for $a^{(m)}_{HSO_4^-}$) with the kinetic of transformation pinonaldehyde $k_{trans}$ inside the particle. Fig. 2 shows $k_{trans}$ (in $s^{-1}$) as a function of $m_{H^+}$ (in mol/kg) and $a^{(m)}_{HSO_4^-}$ (in mol/kg).

Two possible parameterizations can therefore be used to take into account the possible transformation of pinonaldehyde in the particle phase: the pH-dependent parameterization and the $HSO_4^-$-dependent parameterization. For the pH-dependent parameterization, the kinetic rate is computed as:

$$k_{trans} = 2.01 \times 10^{-7} \times exp(0.297 m_{H^+}) \tag{13}$$

whereas the $HSO_4^-$-dependent parameterization is computed with:

$$k_{trans} = 1.53 \times 10^{-7} a^{(m)}_{HSO_4^-} \tag{14}$$

In that case, we assumed that the reaction leads to the formation of the organosulfate formed from pinonaldehyde. The reaction is assumed to be complete as according to the parameterization of Pun and Seigneur (2007), the uptake of pinonaldehyde onto an acidic aerosol should be to be an irreversible process.

    The pH-dependent parameterization (Eq. 13) could be representative of an oligomerization mechanism catalyzed by $H^+$ ion. 15 However, in that case, the uptake of pinonaldehyde should be seen as a reversible pathway which also depends on the humidity of the experiments. The $HSO_4^-$-dependent parameterization (Eq. 14) could be representative of organosulfate formation and therefore pinonaldehyde should in that case act as a sink for sulfates due to the reaction between sulfates and pinonaldehyde.

## 2.3   Aging mechanism

To test the influence of aging on SOA formation, an aging mechanism was developed. The aging of BiA0D is based on the results of Chacon-Madrid and Donahue (2013) who studied SOA formation from pinonaldehyde oxidation. SOA formation from BiA1D is based on the SOA yield from the oxidation of pinonic acid as measured by Müller et al. (2012). For the aging of BiA2D, the parameterizations of Jathar et al. (2015) were used to determine the amount of functionalization and the decrease of volatility due to aging. Nopinone, which is formed from the oxidation of $\beta$-pinene was shown to form a significant amount 25 of SOA and low-volatility products (Sato et al., 2016) and was included in the mechanism. SOA yields from nopinone oxidation was based on Mutzel et al. (2016). The yields of formation of nopinone were based on Hakola et al. (1994) for the reaction of $\beta$-pinene with OH and $O_3$ and on Hallquist et al. (1999) for the reaction of $\beta$-pinene with $NO_3$. Kinetics for aging were taken from the Master Chemical Mechanism v3.3.1 (Jenkin et al., 1997; Saunders et al., 2003).

    For the aging of aromatic SOA compounds, the parameterizations of Jathar et al. (2015) were used to determine the amount 30 of functionalization and the decrease of volatility due to aging. According to this study, oxidation of aromatics should mainly lead to the addition of one oxygen atom. For simplification purposes, we assumed that aging of aromatic SOA leads to the addition a single hydroxy group.

Table 7 shows the aging mechanism used in this study.

This simple mechanism is used here in order to be able to assess the capacity to form oligomers and of the reactive uptake of pinonaldehyde in the long-term formation of SOA.

## 2.4 Long-term SOA formation simulations

The experimental conditions (low relative humidity, oxidation over a few hours, high organic mass loading) are probably very different from typical ambient conditions. To evaluate more precisely the potential effect of oligomerization, SOA formation from the oxidation of $\alpha$-pinene, toluene and TMB over 3 days of evolution with an organic mass loading of 5 $\mu$g m$^{-3}$ was simulated for various humidities with or without oligomerization. The simulated effect of oligomerization is compared to the simulated effect of aging on SOA formation. Octane and NO$_x$ concentrations are set in order to reproduce a level of OH concentrations similar to summer conditions (around 0.001 ppb during daytime). The diurnal cycle was simulated by computing the evolution of the zenith angle at Valencia as a function of local time. This diurnal profile was simulated to take into account that oxidation and aging slow down during night (due to the low concentrations of radicals) whereas oligomerization continues. On the other hand, the relative humidity was assumed constant (whereas under ambient conditions relative humidity probably has a diurnal profile). The structures of pinonic acid for simulations with $\alpha$-pinene and of AnRP2 for simulations with toluene and TMB oxidation are used for the structure of the organic loading. Those simulations are not representative of atmospheric conditions but can be used to illustrate the effect of long-term oligomerization on SOA formation. For simulations with oligomerization, the fraction of monomers inside the aerosol is assumed to be one, i.e., the aerosol is assumed to be mainly constituted by monomers that can react with absorbed compounds. These simulations provide therefore information on the maximal effect that oligomerization can have on SOA yields.

## 3 Results

The simulated degradation of precursors is illustrated in Figures S1 and S2 in supplementary materials. Whereas the mechanism was able to reproduce the degradation of VOC during the beginning of the experiments, medium-term oxidation of toluene seems to be underestimated for some experiments (A1, A3, A4 and A7). Due to this, medium-term formation of toluene SOA may be underestimated.

## 3.1 Comparison with measurements

To ensure of the satisfactory performances of the developed SOA mechanism under uncontrolled conditions, the results of the SOA mechanism are compared to the measurements carried out inside Euphore. Figures 3 and 4 show the results of the model for the biogenic experiments without SO$_2$ and the anthropogenic experiments respectively. Simulated SOA composition for simulations considering oligomerization are illustrated in Supplementary Materials in Figures S3 (biogenic experiments) and S4 (anthropogenic experiments).

Usually, in environmental chamber studies, the wall deposition rate is based on the evolution of concentrations during the last hours of experiments. However, at the last stage of the experiments, oligomerization could still occur and affect the evolution of concentrations and therefore affect the estimation of the wall deposition rate. The estimated deposition rate could be biased if the evolution of concentrations due to oligomerization at the end of the experiment is significant compared to the deposition rate. In these simulations, the wall deposition rate was constrained to reproduce with the model the decrease of SOA volume concentrations (measured with the SMPS) during the last hours of the experiments. Because the computed evolution of SOA concentrations during the last hours can be slightly different with or without oligomerization, the wall deposition rates used with and without oligomerization are different. To examine the effect of the wall deposition rate, the formation of SOA with oligomerization was also simulated with the deposition rate as computed by the simulation without oligomerization to ensure that effect of changes on deposition remain low (changes of concentrations of a few percent) and that hypothesis on oligomerization does not affect the estimated deposition rate.

For the biogenic experiments, the model gives good results (bias lower than 20%) with or without oligomerization for all experiments with slightly better results without oligomerization for experiment B5 and slightly better results with oligomerization for experiment B1. The experiments B2, B3, B4 and B5 have previously been modeled by Santiago et al. (2012a) using a box model version of two air quality models: CMAQ (Carlton et al., 2010) and Chimere (Menut et al., 2013). The authors found a significant overestimation of modeled SOA which was not found with the model used in this study.

For the anthropogenic experiments, the model gives satisfactory results (bias lower than 20%)) with or without oligomerization for experiment A1, A3, A6 and A7 but overestimates concentrations by 30-40% for A2 and A5 and underestimates concentrations by 25% for A8 and A9.

Concentrations for experiment A4 are overestimated by a factor 2. However, experimental conditions for experiment A4 are close to those of experiment A1, except that HONO concentrations are two times higher than for experiment A4. The model gives similar results for A1 (for which the model gives satisfactory results) and A4. It could not explain why concentrations were so different between these two experiments. One possibility is that anthropogenic SOA formation is inhibited at high HONO concentrations.

The alternation of overestimation and underestimation events could be related to differences in chemical regimes. To compare the chemical regimes of the experiments, a ratio representative of the chemical regime (called hereafter "chemical regime ratio") was computed with the sum of VOC concentrations multiplied by their reactivity with OH divided by $NO_x$ concentrations:

$$C_r = \frac{\sum_i k_{OH,i} C_i}{C_{NO_x}} \times 10^{10} \tag{15}$$

with $C_r$ the chemical regime ratio, $C_i$ the concentration in ppb of each VOC and $C_{NO_x}$ the concentration in ppb of $NO_x$, $k_{OH,i}$ the reactivity with OH in molecule$^{-1}$ cm$^3$ s$^{-1}$. $10^{10}$ is a factor set so that the values are close to unity. The chemical regime ratio was used instead of the VOC/$NO_x$ because in this study a mixture of VOC (and not a single VOC) was present in the chamber. The chemical regime ratio takes into account the reactivity of the compounds and can therefore be used to compare different experiments with different mixtures of VOC.

It appears that depending on the chemical regime ratio, the model may overestimate or underestimate the concentrations. Indeed, for the experiments with a low chemical regime ratio (experiments A8 and A9 which have ratios of 0.13 and 0.7 respectively) SOA concentrations are underestimated. On the contrary, SOA concentrations experiments with high chemical regime ratio (experiments A2 and A5 with respectively ratios equal to 2.2 and 1.64) are overestimated. Except for A4, the model gives satisfactory results for the other experiments (having chemical regime ratios between 0.77 and 1.05). It may indicate a more complex $NO_x$ dependency than what is represented in the model.

## 3.2 Effect of oligomerization on SOA formation

For the biogenic and anthropogenic experiments, a low effect of oligomerization on SOA formation was simulated by the "reduced" parameterization (Eq. 3) as shown by Figures 3 and 4. Concentrations are indeed similar with and without oligomerization with low differences compared to the amount of SOA. Although the amount of SOA does not significantly change, the composition of SOA is strongly affected by oligomerization. Indeed, the oligomer content (in mass) at the end of the experiment varies from 68% to 78% for the biogenic experiments and from 38% to 58% for the anthropogenic experiments with a similar range of humidity (between 0.4% and 37%). For a 55% relative humidity, Gao et al. (2004a) determined for $\alpha$-pinene that the oligomer content of SOA could be well over 50%. This result is consistent with the results of our parameterization as an oligomer content around 60% was simulated for such an humidity. On the contrary, for the biogenic experiments, the first order complete oligomerization reaction of Carlton et al. (2010) with a kinetic rate parameter of $9.6 \times 10^{-6}$ s$^{-1}$ (halftime of 20h) simulates low amount of oligomers in SOA (below 15%). Even with the kinetic rate parameter of $3.85 \times 10^{-5}$ s$^{-1}$ (kinetic estimated from the remaining fraction at 100°C measured by Kalberer et al. (2006)), the amount of oligomers estimated (between 29% and 40%) is below the amount estimated by the "reduced" parameterization (Eq. 3). These results indicate that even though the second order "reduced" parameterization (Eq. 3) is not complete, it leads to a faster formation of oligomers and therefore, the parameterization of Carlton et al. (2010) may underestimate the short-term formation of oligomers. On the other hand, as the parameterization of Carlton et al. (2010) is complete, it may also lead to an overestimation of long-term formation of oligomers.

The results of long-term SOA formation simulations are shown in Figures 5, 6 and 7. Without taking into account aging, oligomerization leads to a significant increase of concentrations after 3 days of evolution even at high humidity. Increase factors due to oligomerization are 2.5 for $\alpha$-pinene, 2.0 for toluene and 6.2 for TMB for a relative humidity of 30% and are 1.8 for $\alpha$-pinene, 1.4 for toluene and 3.3 for TMB for a relative humidity of 70%. Oligomerization is therefore a process that could significantly affect long term SOA formation.

Assuming aging leads to a slight decrease of SOA mass for toluene SOA due to fragmentation while it leads to an increase of concentrations for TMB and $\alpha$-pinene SOA due to functionalization. However, the effect of aging on SOA formation simulated here seems less important than the effect of oligomerization.

The different parameterizations of oligomerization are also compared in Fig. 5: the equilibrium second order reaction parameterization developed in this study, the first order complete reaction of Carlton et al. (2010) with a kinetic rate parameter of $9.6 \times 10^{-6}$ s$^{-1}$ (halftime of 20h) and a kinetic rate parameter of $3.85 \times 10^{-5}$ s$^{-1}$ (halftime of 5h). The three different

parameterizations have different impact on SOA formation. The first order parameterization with a kinetic rate parameter of $9.6 \times 10^{-6}$ s$^{-1}$ has a low impact on SOA yields except after a few days of oligomerization. With the kinetic rate parameter of $3.85 \times 10^{-5}$ s$^{-1}$, the increase of SOA yields can be significant after 10 hours. For RH=70% the SOA yield increases by 72% after one day. The second order parameterization gives a faster SOA production. Therefore, oligomerization may be a faster process than simulated in previous modeling studies. However, this figure shows the maximal effect of the second-order parameterization (molar fraction of oligomers of 1). Depending on conditions, the second order parameterization may only have a low impact on SOA concentrations and the formation of oligomers could rapidly reach an equilibrium.

Depending on the conditions, oligomerization could also lead to a decrease of SOA concentrations as the partitioning is inversely proportional to the mean molar mass of organic aerosols (Pankow, 1994), which will increase with the oligomer formation. Increasing the mean molar mass by a factor 2 leads to a decrease around 40% of the SOA formed without oligomerization, indicating that the partitioning of monomers is sensitive to the value of the mean molar mass.

### 3.3  Uptake of pinonaldehyde onto acidic aerosols

The pH (Eq. 13) and HSO$_4^-$ (Eq. 14) parameterizations and the parameterization of Pun and Seigneur (2007) (Eq. 10) assuming equilibrium are compared and evaluated against experiments B6 and B7 where $\alpha$-pinene and limonene are oxidized in the presence of SO$_2$. The oxidation of SO$_2$ leads to the formation of sulfuric acid and therefore to acidic aerosols. The amount of SOA formed with each parameterization is compared to the results of the experiments (Fig. 8).

The parameterization of Pun and Seigneur (2007) (Eq. 10) leads to a significant overestimation of SOA concentrations. With this parameterization, BiA0D is entirely absorbed by the acidic aerosol. On the contrary, using the pH (Eq. 13) and HSO$_4^-$ (Eq. 14) parameterizations which take into account the dynamic of the uptake, no significant formation of SOA is formed by this pathway. The two parameterizations give almost the same results as assuming no uptake. These results indicate that the dynamic of the uptake of pinonaldehyde need to be taken into account.

Pinonaldehyde was found to be too volatile to form significant SOA by this pathway. The long-term formation of SOA by this pathway is tested with the simulation described in the previous section but in the presence of 2 $\mu$g/m$^3$ of sulfuric acid. Fig. 5 shows that even at low humidity (RH=30%), the amount of SOA formed is not significant after 3 days of evolution. Therefore, SOA formation from the reactive uptake by acidic aerosol of pinonaldehyde probably does not contribute significantly to SOA formation.

However, it could be possible that aldehyde compounds less volatile than pinonaldehyde react and form significant amount of SOA by this pathway. To test this hypothesis, we assumed that organosulfate could be formed from the compounds AnRP1 and AnRP2 (which have an aldehyde group in their molecular surrogate structure) using the HSO$_4$ parameterization (Eq. 14). The kinetic rate parameter should depend on the compound but the HSO$_4$ parameterization (Eq. 13) probably provides a good estimation of this phenomenon. Fig. 9 shows the amount of organosulfate that would be formed by this pathway for a humidity of 70% from the oxidation of toluene. Long-term organosulfate formation seems possible by this pathway (even at high humidity) as significant amount of organosulfate (13 to 18% of SOA) is formed with this assumption and that a significant mass of aldehydes has been converted into organosulfates. However, if high concentrations of ammonia are present in the atmosphere,

pH will increase and $HSO_4^-$ will decrease. In case of high concentrations of ammonia, the kinetic rate could be too low for this process to be significant. This process would need for sulfate to not be fully neutralized by ammonia.

## 3.4 Investigation of the effect of particle-phase diffusion and wall losses of organic vapors

To evaluate the effect of the particle phase viscosity on SOA formation, the SOAP model was run for a particle-phase diffusion coefficient of $2 \times 10^{-16}$ molecules/cm$^2$/s which correspond to the order of magnitude of values determined for toluene SOA at low humidities (Song et al., 2016).

To investigate wall losses of organic vapors, wall losses were treated as in Zhang et al. (2014) by using a first-order wall loss rate and the organic aerosol equivalent wall (set to 10 mg/m$^3$ following Zhang et al. (2014)). According to McMurry and Grosjean (1985), the wall vapor loss rate $k_w$ can be computed with the following formula:

$$k_w = \frac{A}{V} \frac{\frac{a_w c}{4}}{1 + \frac{\pi}{2} \frac{\frac{a_w c}{4}}{\sqrt{k_e D_{gas}}}} \tag{16}$$

with A/V the surface on volume ratio of the chamber (equal to 1 m$^{-1}$ for EUPHORE), $a_w$ the mass accommodation coefficient of organic vapors onto the wall, c is the mean thermal speed of the molecules, $k_e$ is the coefficient of eddy diffusion, $D_{gas}$ is the molecular diffusivity.

As the mixing is carried out by two fans with a flux of 4000 m$^3$/h, the characteristic time for the mixing inside the chamber is estimated to be around 90 s for EUPHORE and $k_w$ is estimated to be around $10^{-4}$ s$^{-1}$. Following Zhang et al. (2014), the kinetic of evaporation of compounds from the wall $k_{w,off}$ is computed with:

$$k_{w,off} = \frac{k_w}{K_{p,w} C_w} \tag{17}$$

with $K_{p,w}$ the gas/wall partition coefficient (chosen equal to the gas/particle partition coefficient) and $C_w$ equivalent wall OA concentration (set to 10 000 $\mu$g m$^{-3}$).

The effects of the particle-phase viscosity and of wall losses of vapor is illustrated in Fig. 10 for experiments A5, A7 and B1. Results for other experiments (biogenic and anthropogenic) are similar to these experiments and are illustrated in Supplementary Materials in figures S5 and S6.

Generally, similar results are obtained between non-viscous and viscous aerosols. Assuming viscous aerosols can lead to an increase of SOA concentrations due to the limitation of evaporation with low diffusion. As found by Couvidat and Sartelet (2015), condensation of low volatility compounds is possible without diffusing into the particle by condensing at the gas/particle interface. The condensation of low volatility compounds can create a layer onto which more volatile compounds can condense to respect Raoult's law at the interface. However, viscosity can prevent the evaporation of those more volatile compounds by preventing their diffusion from the core of the particle to the interface. For non-viscous aerosols, deposition of particles to the walls lead to a decrease of the absorbing mass (mass of the organic aerosol). As the gas/particle partitioning depends on the absorbing mass, SVOC will evaporate to maintain the gas/particle partitioning whereas this evaporation will be limited for a viscous aerosol.

Assuming viscous aerosols can also lead to a decrease of SOA sensitivity to changes of conditions during the experiments. For example, experiment A5 is characterized by a decrease of the temperature during the experiment leading to a decrease of volatility. Whereas concentrations increase when assuming non-viscous aerosols, concentrations seem unaffected by the change of temperature when assuming viscous aerosols (as compounds are absorbed by the core of particles with a slow kinetic rate). The shape of modeled SOA concentration curve is closer to measurement when assuming viscous aerosols.

Taking into account wall losses of vapors leads to a decrease of SOA concentrations (as illustrated in Fig. 10), particularly for the anthropogenic experiments (decrease between 8 and 30%) due to the lower organic aerosol loading compared to the biogenic experiments. As the developed mechanism is based on the methodology of Odum et al. (1996) by using experimental results from Teflon chambers, it could be possible that the parameters of the mechanism (saturation vapor pressures and stoichiometric coefficients for the formation of SVOC) are biased. As stronger organic aerosol loading favors condensation onto organic aerosol over condensation onto walls, wall losses of vapor may produce a shift in apparent volatilities; the organic aerosol formed in chambers may appear more volatile than it should be. For both the biogenic and anthropogenic experiments, results can be corrected by decreasing the saturation vapor pressures by 20% as better or similar results could be obtained than without taking into account wall losses of organic vapors. These results indicate that volatility determined by the use of Odum's curves may be slightly overestimated. However, wall deposition of organic vapors could also lead to an underestimation in the SOA mechanism of stoichiometric coefficients.

## 4   Conclusions

Several parameterizations were developed in this study. First, in the spirit of updating the $H^2O$ mechanism (Couvidat et al., 2012), a new mechanism was developed to take into account hydrophilic and hydrophobic properties of monoterpene and aromatic SOA and introduce aging in the mechanism. The performance of the new mechanism was evaluated by comparison to experimental results from previous studies carried out in the EUPHORE chamber in Valencia. Second, parameterizations to take into account oligomerization and the uptake of pinonaldehyde onto acidic aerosols were developed. Finally, the effects of particle viscosity and wall deposition of vapors were investigated.

Without taking into account wall losses of vapors, satisfactory results are obtained for the biogenic experiments. However, the experiments used for the comparison present a high organic aerosol loading (between 50 and 150 $\mu$g/m$^3$), much higher than the values normally observed under atmospheric conditions. More experiments at low organic aerosol loading should be carried out to provide a wider observational dataset that could be used to evaluate model developments. A good performance is also found for most of the anthropogenic experiments. However, for the experiments with the lower chemical regime ratio (Eq. 15, product of VOC concentrations and their reactivity divided by $NO_x$ concentrations), the model underestimates concentrations by 30 % whereas, for the experiments with the higher chemical regime ratio, the model overestimates concentrations by 30 %. This could indicate a more complex $NO_x$ chemistry than the one taken into account in the model. However, this discrepancy could be due to the difficulty to simulate properly radicals inside the chamber under some conditions. More experiments should

be carried out to confirm these results and improve the $NO_x$ dependency inside the model.

Contrary to Carlton et al. (2010) who represented oligomerization as a first-order irreversible process, oligomerization was represented in the present study (as an effort to provide a more realistic representation) as a second-order reversible reaction unfavored by humid conditions by analogy to oligomerization reactions expected to occur in the particles, such as esterification, hemiacetalization, aldolization, peroxyhemiacetalization. With this new parameterization, oligomerization was shown to have little impact on SOA mass during the experiments. However, oligomerization was found to significantly affect SOA composition as a large part of SOA is constituted (more than 50% for the biogenic experiments) by oligomers according to the simulations. These results seem to be consistent with the high amount of oligomers reported by Gao et al. (2004a). On the opposite, a low amount of oligomers was simulated with the parameterization of Carlton et al. (2010) (below 15% at the end of the experiments). The results of the present study with the new parameterization indicate that oligomerization may be faster than simulated by the parameterization of Carlton et al. (2010) but as the same time may reach an equilibrium. However, more efforts should be deployed to improve this parameterization. Indeed, this parameterization represents a "bulk oligomerization" and does not account for differences in the reactivity of monomers. This parameterization should nonetheless be deployed and tested in 3D air quality models to diagnose the effect of oligomerization on SOA formation.

Instead of representing the uptake of pinonaldehyde by an equilibrium phenomena (like the parameterization of Pun and Seigneur (2007)), we chose to take into account the kinetic of this uptake, which was shown to limit strongly the uptake of pinonaldehyde Liggio and Li (2006b). The uptake of pinonaldehyde was found to be too slow to contribute significantly to the formation of organic aerosols, suggesting that the approach of Pun and Seigneur (2007) (Eq. 10) greatly overestimates the effect of the particle acidity on SOA formation. However, it could be possible that the uptake of aldehydes less volatile than pinonaldehyde (or more reactive) is strongly influenced by the particle acidity leading to the formation of oligomers or organosulfate.

Taking into account wall losses of organic vapors with the parameters of Zhang et al. (2014) led to a decrease of SOA concentrations that up to 30%. As the mechanism developed in the present study is based on the methodology of Odum et al. (1996) by using experimental results from Teflon chambers, it the parameters of the mechanism (saturation vapor pressures and stoichiometric coefficients for the formation of SVOC) could be biased. The simulated decrease of SOA concentrations could however be compensated by decreasing the volatility of SVOC by 20% (decrease of the value of saturation vapor pressures by 20%).

Regarding RH, the model indicates that RH may strongly influence SOA yields which is consistent with the results of Healy et al. (2009). Unfortunately, the conditions of the experiments used in this study do not cover the full range of RH. More experiments carried out at high humidity could be used to evaluate the model performance for other humidities.

Finally, these parameterizations should be implemented in a 3D air quality models to evaluate their impact on SOA formation in the atmosphere. Oligomerization should especially be studied into detail as it could lead either to an increase or a decrease (by increasing the mean molar mass that would lead to a decrease of the partitioning constant of monomers) of SOA concentrations.

35  *Acknowledgements.* This work was funded by the French Ministry in charge of ecology.

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

**Table 1.** Initial concentrations in ppb, temperature (T) in Kelvin and relative humidity in % for biogenic experiments.

| Experiment | Isoprene | $\alpha$-pinene | Limonene | NO | NO$_2$ | HONO | SO$_2$ | T | RH |
|---|---|---|---|---|---|---|---|---|---|
| B1 | 107 | 66 | 58 | 34 | 128 | 99 | 0 | 302-307 | 0.5-3 |
| B2 | 92 | 50 | 50 | 48 | 0 | 87 | 0 | 298-300 | 30-26 |
| B3 | 122 | 71 | 40 | 41 | 0 | 53 | 0 | 297-300 | 19-22 |
| B4 | 0 | 63 | 65 | 32 | 0 | 101 | 0 | 294-298 | 8-13 |
| B5 | 99 | 59 | 53 | 150 | 0 | 307 | 0 | 295-297 | 8-11 |
| B6 | 87 | 50 | 51 | 244 | 89 | 40 | 513 | 295-300 | 15-19 |
| B7 | 55 | 79 | 76 | 198 | 0 | 165 | 461 | 302-305 | 20-30 |

**Table 2.** Initial concentrations in ppb, temperature (T) in Kelvin and relative humidity in % for anthropogenic experiments. SO$_2$ was not present for those experiments.

| Experiment | Toluene | o-xylene | TMB | Octane | NO | NO$_2$ | HONO | T | RH |
|---|---|---|---|---|---|---|---|---|---|
| A1 | 102 | 22 | 153 | 85 | 19 | 0 | 99 | 299-305 | 10-16 |
| A2 | 200 | 49 | 300 | 155 | 23 | 0 | 75 | 302-305 | 9-18 |
| A3 | 48 | 11 | 106 | 42 | 23 | 0 | 71 | 302-307 | 6-14 |
| A4 | 98 | 24 | 160 | 79 | 37 | 0 | 156 | 297-307 | 6-13 |
| A5 | 97 | 21 | 146 | 81 | 4 | 8 | 52 | 297-308 | 7-14 |
| A6 | 93 | 22 | 146 | 78 | 21 | 0 | 94 | 300-308 | 0.4 |
| A7 | 107 | 26 | 160 | 89 | 21 | 0 | 89 | 306-309 | 7-10 |
| A8 | 116 | 29 | 19 | 10 | 57 | 0 | 119 | 302-305 | 15-18 |
| A9 | 81 | 21 | 118 | 65 | 31 | 0 | 90 | 299-303 | 28-37 |

**Table 3.** Properties of the different surrogate SOA species.

| Surrogate | Molecular structure | MW [a] | $P^0$ [b] | $\Delta H_{vap}$ [c] |
|---|---|---|---|---|
| AnRP1 |  | 202 | $1.01 \times 10^{-7}$ | 50 |
| AnRP2 |  | 160 | $2.68 \times 10^{-7}$ | 50 |
| AnIP1 |  | 132 | $2.36 \times 10^{-6}$ | 50 |
| AnIP2 |  | 167 | $1.24 \times 10^{-5}$ | 50 |
| AnPER |  | 190 | non-volatile | - |
| AnRP1a | AnRP1 + 1 group OH | 218 | non-volatile | - |
| AnRP2a | AnRP2 + 1 group OH | 176 | non-volatile | - |
| AnIP1a | AnIP1 + 1 group OH | 148 | $1.36 \times 10^{-7}$ | 50 |
| AnIP2a | AnIP2 + 1 group OH | 183 | $2.58 \times 10^{-7}$ | 50 |
| AnIP2b | AnIP2a + 1 group OH | 199 | $5.39 \times 10^{-9}$ | 50 |
| BiA0D | pinonaldehyde | 168 | $1.0 \times 10^{-3}$ | 50 |
| BiA1D | pinonic acid | 184 | $5.61 \times 10^{-5}$ | 50 |
| BiA2D | pinic acid | 186 | $1.67 \times 10^{-6}$ | 50 |
| BiA3D | 3-methyl-1,2,3-butanetricarboxylic acid | 202 | non-volatile | - |

[a] Molecular weight [g.mol$^{-1}$]

[b] Saturation vapor pressure [torr] at 298K

[c] Enthalpy of vaporization [kJ.mol$^{-1}$]

**Table 4.** Reactions leading to SOA formation[a] from toluene (referred as TOL).

| Reaction | Kinetic rate parameter (molecule$^{-1}$.cm$^3$.s$^{-1}$) |
|---|---|
| TOL + OH → ... + 0.25 TOLP + OH | $1.80 \times 10^{-12} \times \exp(\frac{355}{T})$ |
| TOLP + HO$_2$ → TOLlowNOx + HO$_2$ | $3.75 \times 10^{-13} \times \exp(\frac{980}{T})$ |
| TOLP + A → TOLhighNOx + A | A = NO: $2.70 \times 10^{-12} \times \exp(\frac{360}{T})$ |
| | A = NO$_3$: $1.2 \times 10^{-12}$ |
| | A = MO$_2$: $3.56 \times 10^{-14} \times \exp(\frac{708}{T})$ |
| | A = ACO$_3$: $7.40 \times 10^{-13} \times \exp(\frac{765}{T})$ |
| TOLlowNOx + OH → TOLlowNOxRAD + OH | $6.90 \times 10^{-11}$ |
| TOLhighNOx + OH → TOLlowNOxRAD + OH | $6.90 \times 10^{-11}$ |
| TOLlowNOxRAD + HO$_2$ → 0.697 AnPER + HO$_2$ | See TOLP + HO$_2$ reaction |
| TOLlowNOxRAD + A → 0.131 AnRP2 + 0.324 AnIP1 + A | See TOLP + A reaction |
| TOLhighNOxRAD + HO$_2$ → 0.131 AnRP2 + 0.324 AnIP1 + HO$_2$ | See TOLP + HO$_2$ reaction |
| TOLhighNOxRAD + A → 0.131 AnRP2 + 0.324 AnIP1 +A | See TOLP + A reaction |

[a] Oxidants may be present as both reactants and products so that a reaction added to RACM2 will not affect the original photochemical oxidant concentrations. MO$_2$ and ACO$_3$ are the methylperoxy radical and the peroxy-acetyl radical respectively. A is either NO, NO$_3$, MO$_2$ or ACO$_3$.

**Table 5.** Reactions leading to SOA formation[a] from o-xylene (referred as XYL) and 1,3,5-trimethylbenzene (referred as TMB).

| Reaction | Kinetic rate parameter (molecule$^{-1}$.cm$^3$.s$^{-1}$) |
|---|---|
| XYL + OH → ... + 0.274 XYLP | $1.70 \times 10^{-11}$ x $\exp(\frac{116}{T})$ |
| XYLP + HO$_2$ → XYLlowNOx + HO$_2$ | See TOLP + HO$_2$ reaction |
| XYLP + A → XYLhighNOx + A | See TOLP + A reaction |
| XYLlowNOx + OH → XYLlowNOxRAD + OH | $6.90 \times 10^{-11}$ |
| XYLhighNOx + OH → XYLlowNOxRAD + OH | $6.90 \times 10^{-11}$ |
| XYLlowNOxRAD + HO$_2$ → 0.611 AnPER + HO$_2$ | See TOLP + HO$_2$ reaction |
| XYLlowNOxRAD + A → 0.0529 AnRP1 + 0.344 AnIP2 + A | See TOLP + A reaction |
| XYLhighNOxRAD + HO$_2$ → 0.0529 AnRP1 + 0.344 AnIP2 + HO$_2$ | See TOLP + HO$_2$ reaction |
| XYLhighNOxRAD + A → 0.0529 AnRP1 + 0.344 AnIP2 + A | See TOLP + A reaction |
| TMB + OH → ... + 0.274 TMBP | $5.67 \times 10^{-11}$ |
| TMBP + HO$_2$ → TMBlowNOx + HO$_2$ | See TOLP + HO$_2$ reaction |
| TMBP + A → TMBhighNOx + A | See TOLP + A reaction |
| TMBlowNOx + OH → TMBlowNOxRAD + OH | $6.90 \times 10^{-11}$ |
| TMBhighNOx + OH → TMBlowNOxRAD + OH | $6.90 \times 10^{-11}$ |
| TMBlowNOxRAD + HO$_2$ → 0.611 AnPER + HO$_2$ | See TOLP + HO$_2$ reaction |
| TMBlowNOxRAD + A → 0.0117 AnRP1 + 0.250 AnIP2 + A | See TOLP + A reaction |
| TMBhighNOxRAD + HO$_2$ → 0.0117 AnRP1 + 0.250 AnIP2 + HO$_2$ | See TOLP + HO$_2$ reaction |
| TMBhighNOxRAD + A → 0.0117 AnRP1 + 0.250 AnIP2 +A | See TOLP + A reaction |

[a] Oxidants may be present as both reactants and products so that a reaction added to RACM2 will not affect the original photochemical oxidant concentrations. MO$_2$ and ACO$_3$ are the methylperoxy radical and the peroxy-acetyl radical respectively. A is either NO, NO$_3$, MO$_2$ or ACO$_3$.

**Table 6.** Reactions leading to SOA formation[a] from $\alpha$-pinene (referred as API), $\beta$-pinene (referred as BPI) and limonene (referred as LIM).

| Reaction | Kinetic rate parameter (molecule$^{-1}$.cm$^3$.s$^{-1}$) |
|---|---|
| API + OH → 0.30 BiA0D + 0.40 BiA2D + OH | $1.21 \times 10^{-11} \times \exp(\frac{440}{T})$ |
| API + O$_3$ → APIO3RAD + O$_3$ | $5.00 \times 10^{-16} \times \exp(\frac{-530}{T})$ |
| APIO3RAD + HO$_2$ → 0.024 BiA3D + 0.15 BiA2D + 0.38 BiA1D + HO$_2$ | $4.10 \times 10^{-13} \times \exp(\frac{790}{T})$ |
| APIO3RAD + NO → 0.085 BiA2D + 0.24 BiA1D + NO | $8.8 \times 10^{-13} \times \exp(\frac{180.2}{T})$ |
| API + NO$_3$ → 0.70 BiA0D + NO$_3$ | $1.19 \times 10^{-12} \times \exp(\frac{-490}{T})$ |
| BPI + OH → 0.07 BiA0D + 0.08 BiA1D + 0.06 BiA2D + 0.27 NOPINONE +OH | $2.38 \times 10^{-11} \times \exp(\frac{357}{T})$ |
| BPI + O$_3$ → 0.09 BiA0D + 0.022 BiA3D + 0.045 BiA2D + 0.20 BiA1D + 0.17 NOPINONE + O$_3$ | $1.50 \times 10^{-17}$ |
| BPI + NO$_3$ → 0.02 BiA0D + 0.21 BiNIT + 0.02 NOPINONE + NO$_3$ | $2.51 \times 10^{-12}$ |
| LIM + OH → 0.35 BiA0D + 0.15 BiA2D + OH | $4.20 \times 10^{-11} \times \exp(\frac{401}{T})$ |
| LIM + O$_3$ → LIMO3RAD + O$_3$ | $2.95 \times 10^{-15} \times \exp(\frac{783}{T})$ |
| LIMO3RAD + HO$_2$ → 0.14 BiA3D + 0.44 BiA2D+ 0.42 BiA1D + HO$_2$ | $4.10 \times 10^{-13} \times \exp(\frac{790}{T})$ |
| LIMO3RAD + NO → 0.14 BiA3D + 0.5 BiA2D + 0.36 BiA1D + NO | $8.8 \times 10^{-13} \times \exp(\frac{180.2}{T})$ |
| LIM + NO$_3$ → 0.69 BiA0D + 0.28 BiNIT + NO$_3$ | $1.22 \times 10^{-11}$ |

[a] Oxidants may be present as both reactants and products so that a reaction added to RACM2 will not affect the original photochemical oxidant concentrations. MO$_2$ and ACO$_3$ are the methylperoxy radical and the peroxy-acetyl radical respectively. A is either NO, NO$_3$, MO$_2$ or ACO$_3$.

**Table 7.** Aging mechanism of SVOCs[a].

| Reaction | Kinetic rate parameter (molecule$^{-1}$.cm$^3$.s$^{-1}$) | References for used SOA yield |
|---|---|---|
| BiA0D + OH → RA0D + OH | $9.0 \times 10^{-12}$ | - |
| RA0D + HO$_2$ → BiA1D + HO$_2$ | $5.20 \times 10^{-13} \times \exp(\frac{980}{T})$ | Chacon-Madrid and Donahue (2013) |
| RA0D + NO → 0.6 BiA1D + 0.075 BiA3D + NO | $7.50 \times 10^{-12} \times \exp(\frac{290}{T})$ | Chacon-Madrid and Donahue (2013) |
| BiA1D + OH → 0.061 BiA3D + OH | $1.12 \times 10^{-11}$ | Müller et al. (2012) |
| BiA2D + OH → 0.4 BiA3D + OH | $7.29 \times 10^{-12}$ | Jathar et al. (2015) |
| NOPINONE + OH → 0.16 BiA3D + OH | $1.55 \times 10^{-11}$ | Mutzel et al. (2016) |
| AnRP1 + OH → 0.26 AnRP1a + OH | $6.0 \times 10^{-12}$ | Jathar et al. (2015) |
| AnRP2 + OH → 0.06 AnRP2a + OH | $6.0 \times 10^{-12}$ | Jathar et al. (2015) |
| AnIP1 + OH → 0.04 AnIP1a + OH | $6.0 \times 10^{-12}$ | Jathar et al. (2015) |
| AnIP1a + OH → Volatile products + OH | $6.0 \times 10^{-12}$ | Jathar et al. (2015) |
| AnIP2 + OH → 0.48 AnIP2a + OH | $6.0 \times 10^{-12}$ | Jathar et al. (2015) |
| AnIP2a + OH → 0.38 AnIP2b + OH | $6.0 \times 10^{-12}$ | Jathar et al. (2015) |

[a] Oxidants may be present as both reactants and products so that a reaction added to RACM2 will not affect the original photochemical oxidant concentrations. MO$_2$ and ACO$_3$ are the methylperoxy radical and the peroxyacetyl radical respectively.

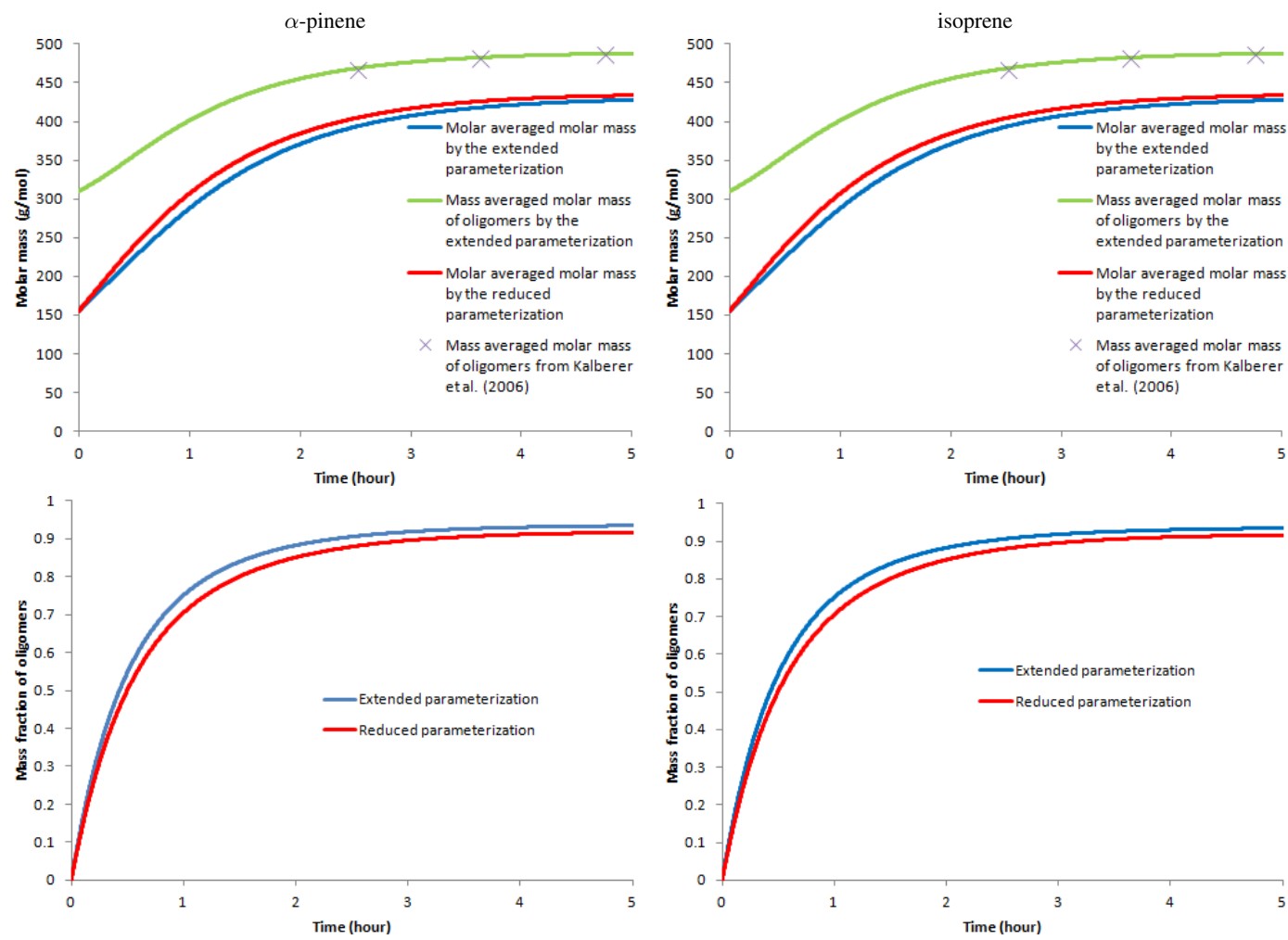

**Figure 1.** Temporal evolution of molar masses (top) and of the mass fraction of oligomer (down) for the extended (Eq. 1) and reduced (Eq. 3) parameterizations.

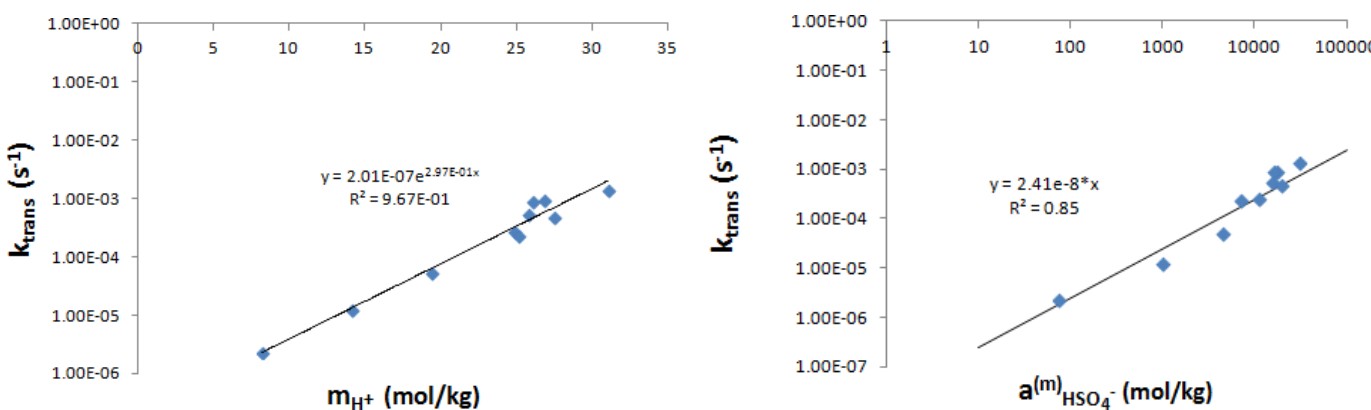

**Figure 2.** Kinetic rate of transformation of pinonaldehyde as a function of $m_{H+}$ the molality of ion $H^+$ (left) and $a^{(m)}_{HSO_4^-}$ the activity on a molality basis of ion $HSO_4^-$ (right).

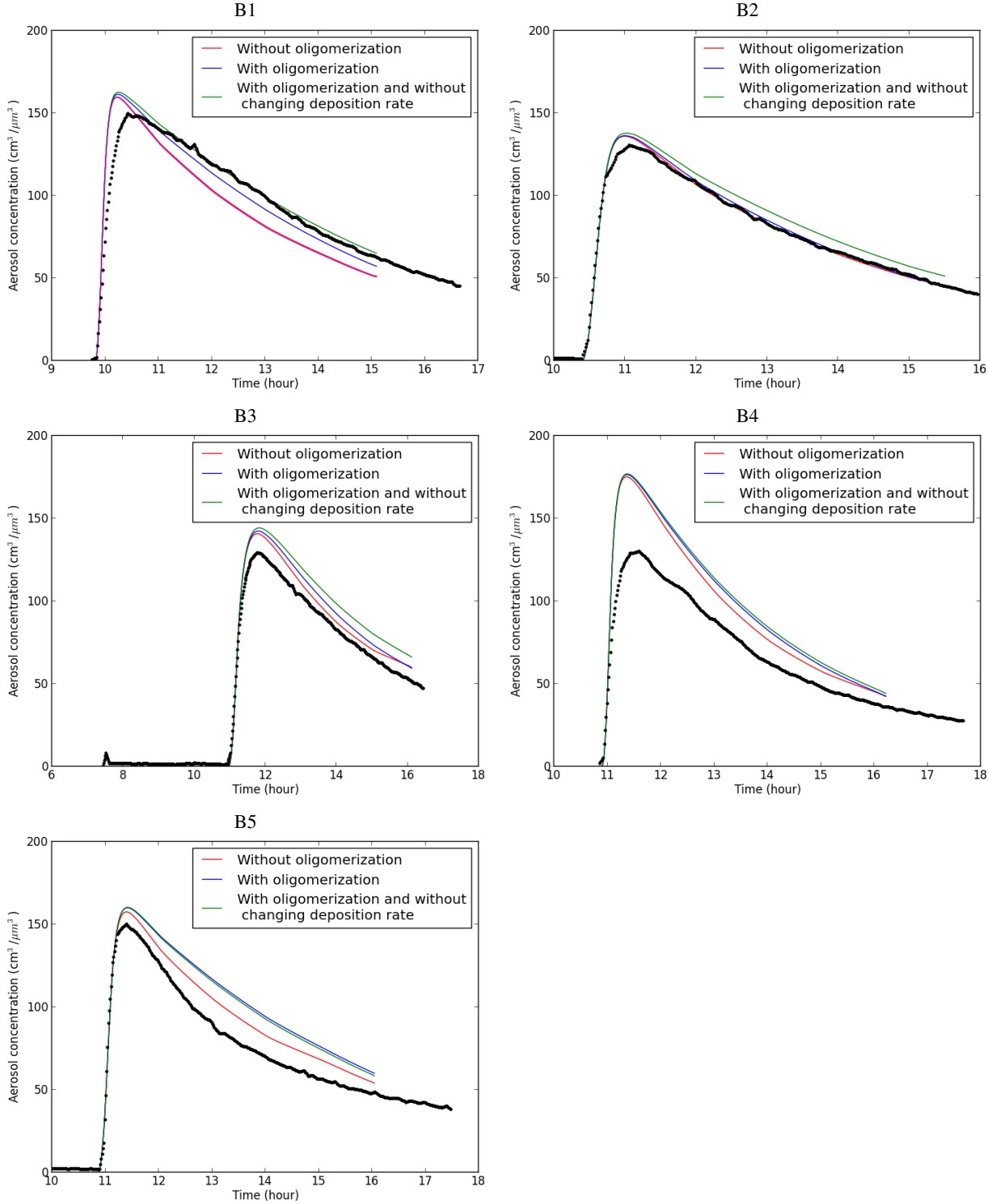

**Figure 3.** Aerosol concentration formation for the biogenic experiments without SO$_2$. Black lines correspond to SMPS measurements.

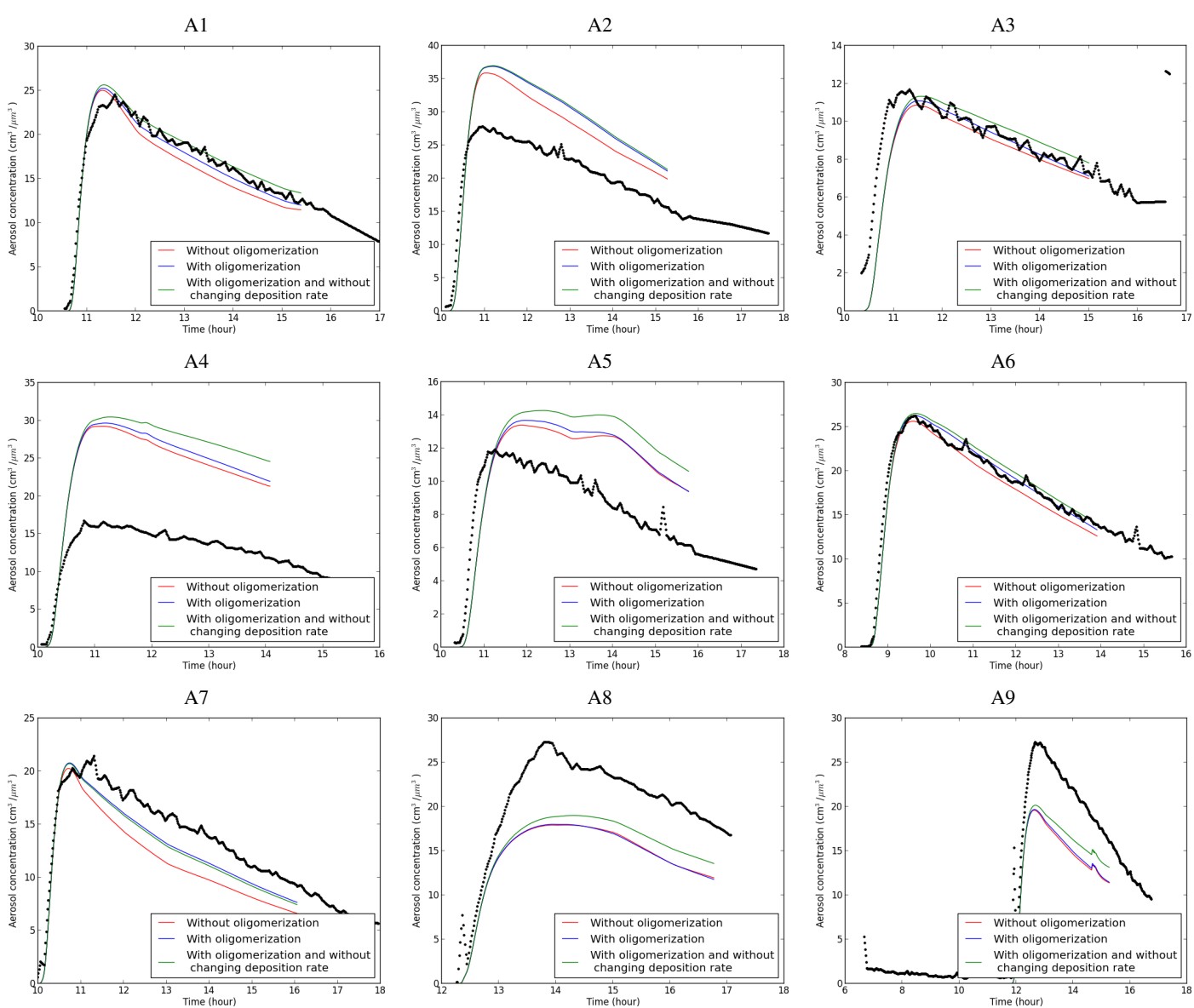

**Figure 4.** Aerosol concentration formation for the anthropogenic experiments. Black lines correspond to SMPS measurements.

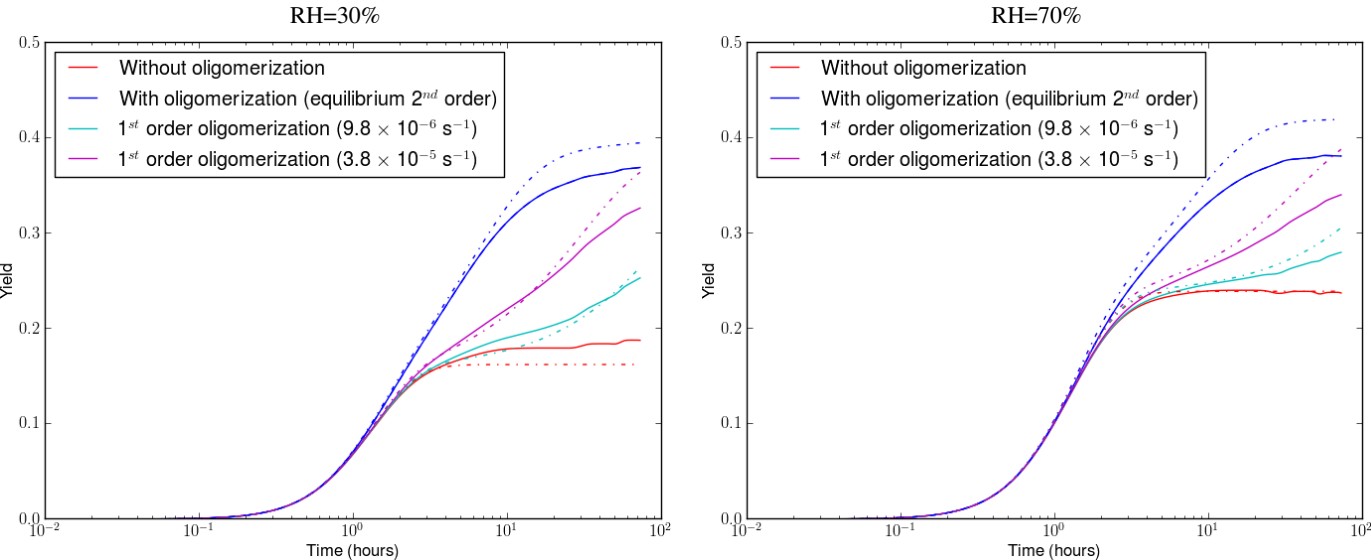

**Figure 5.** Evolution of the SOA yield from $\alpha$-pinene oxidation as a function of time for an organic mass loading of 5 $\mu$g m$^{-3}$. Solid lines correspond to SOA formation with aging. Dashed lines (-.) correspond to SOA formation without aging.

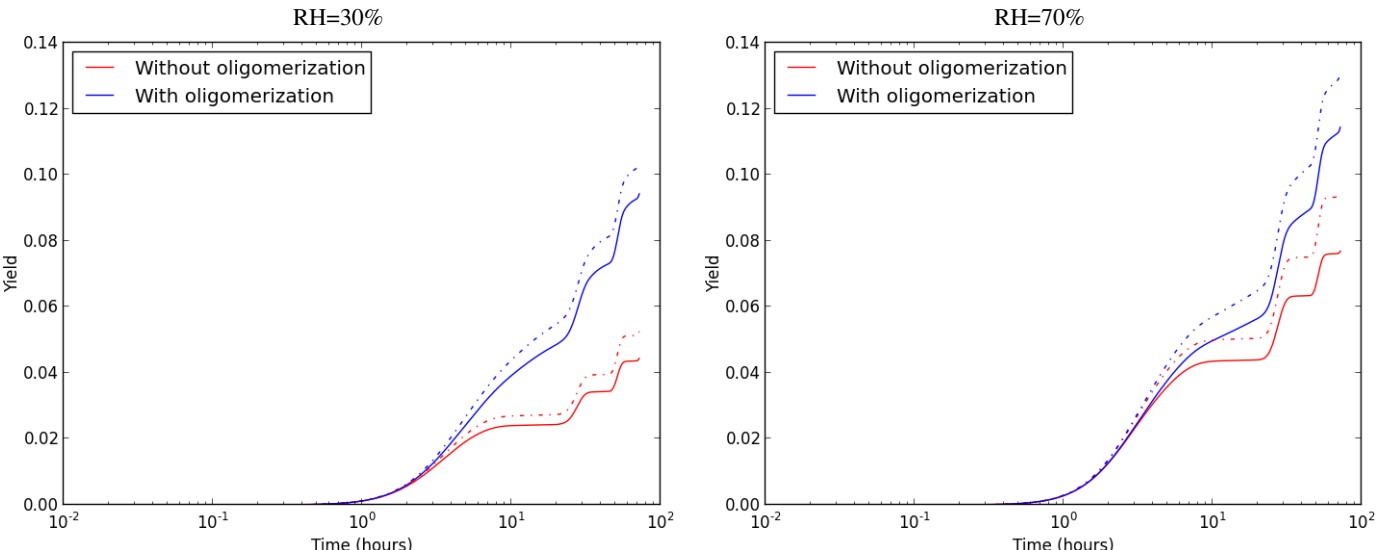

**Figure 6.** Evolution of the SOA yield from toluene oxidation as a function of time for an organic mass loading of 5 $\mu$g m$^{-3}$. Solid lines correspond to SOA formation with aging. Dashed lines correspond to SOA formation without aging.

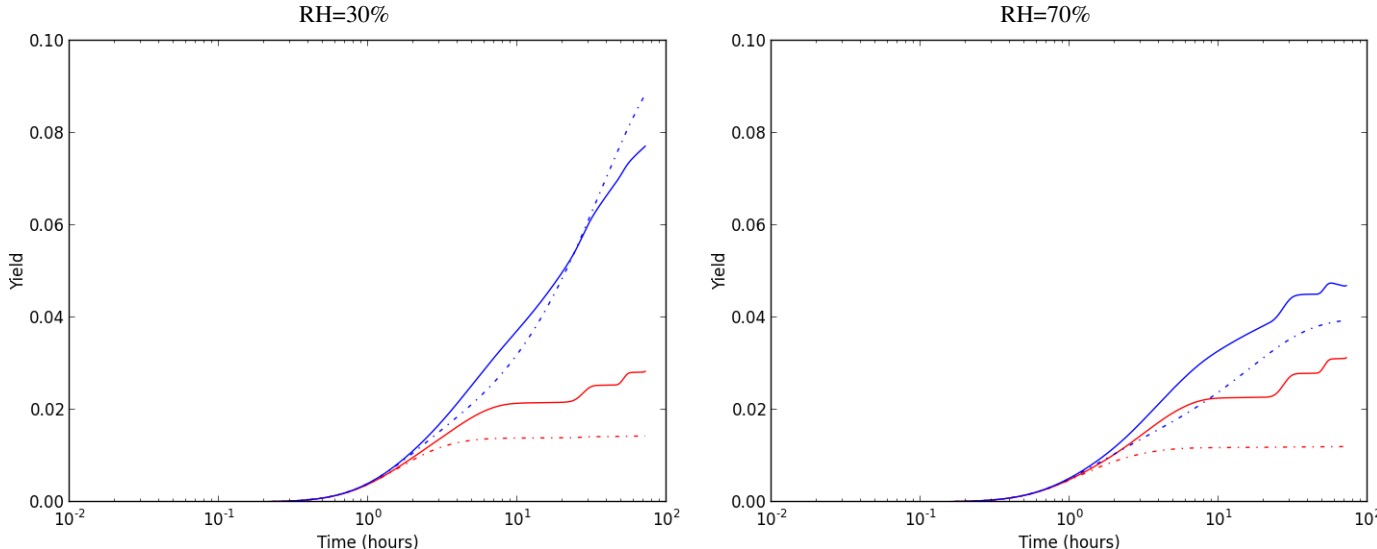

**Figure 7.** Evolution of the SOA yield from trimethylbenzene oxidation as a function of time for an organic mass loading of 5 $\mu$g m$^{-3}$. Solid lines correspond to SOA formation with aging. Dashed lines correspond to SOA formation without aging.

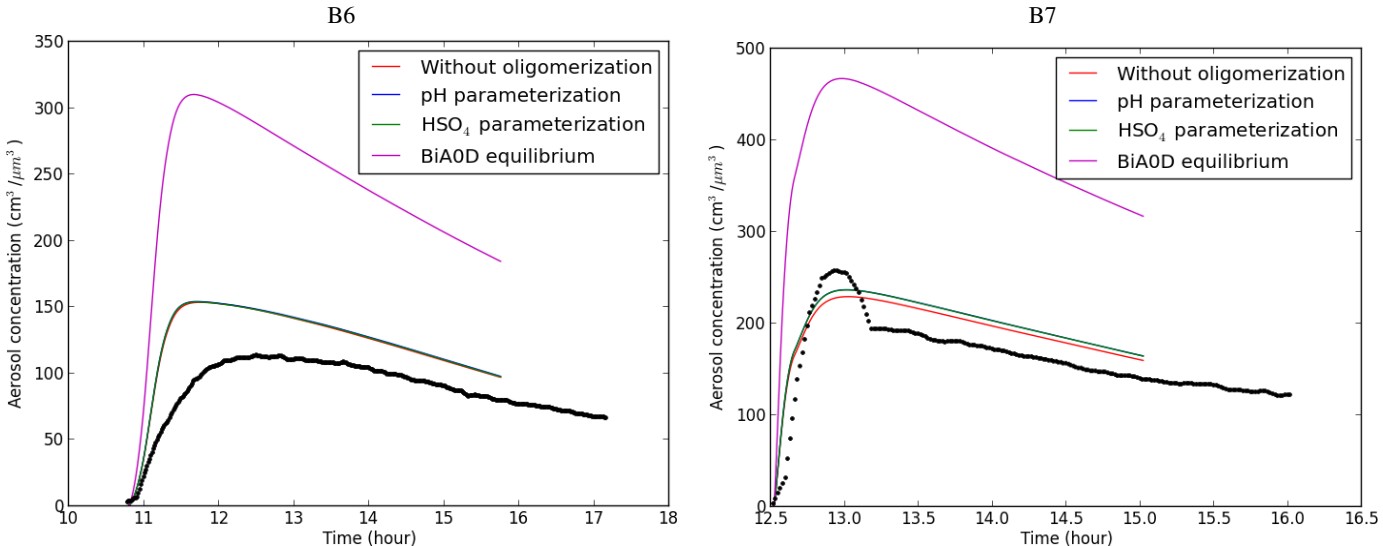

**Figure 8.** Aerosol concentration formation for the biogenic experiments with SO$_2$. Black lines correspond to SMPS measurements.

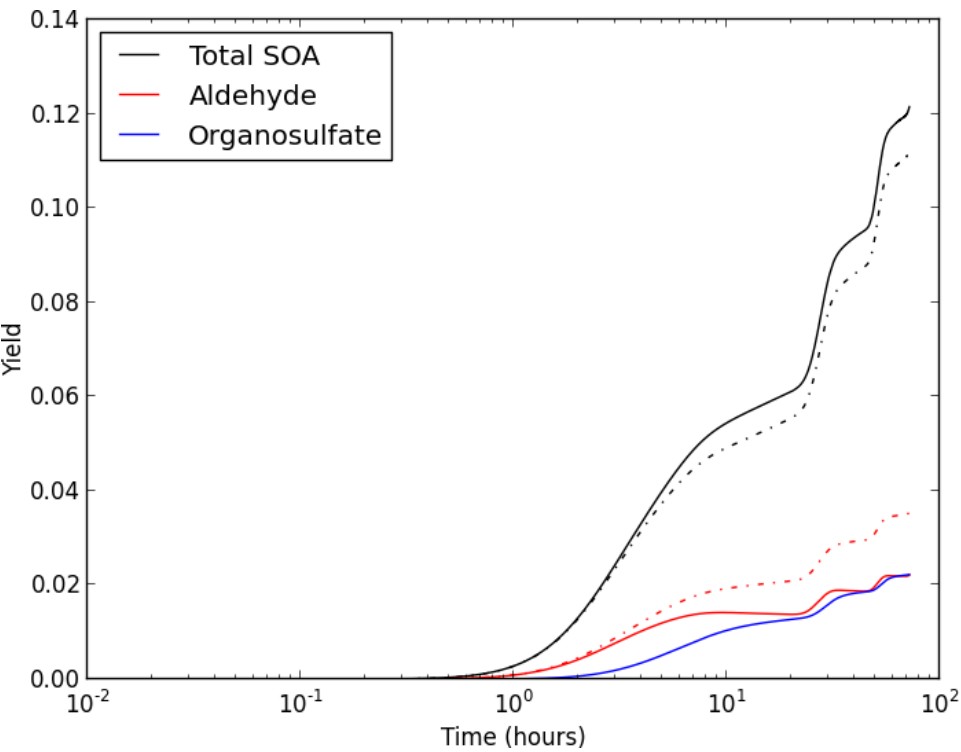

**Figure 9.** Yield of formation of SOA, aldehydes and organosulfates with (solid lines) or without (dashed lines) the conversion of aldehydes into organosulfates as a function of time for an organic mass loading of 5 $\mu$g m$^{-3}$ with 2 $\mu$g m$^{-3}$ of sulfates.

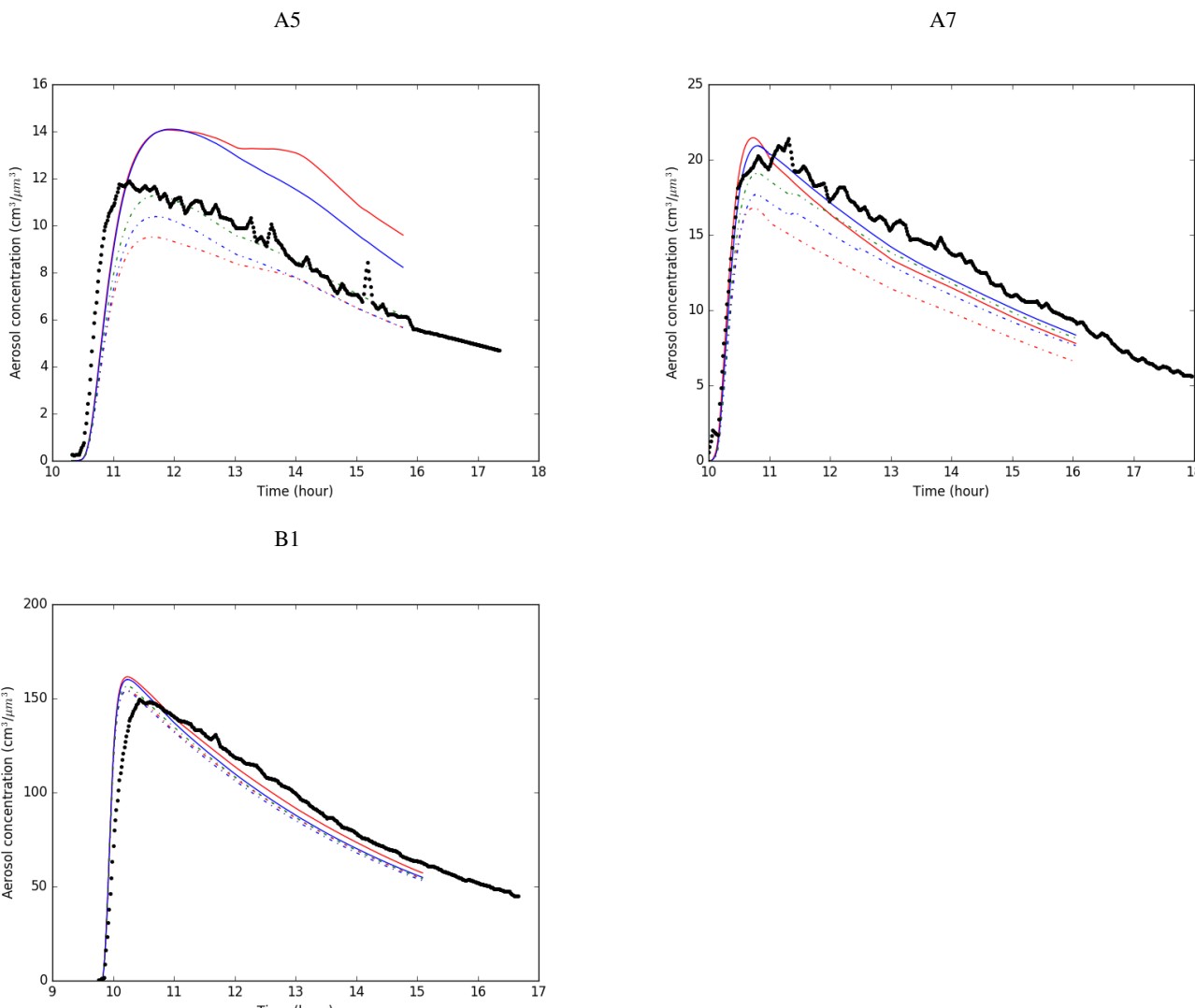

**Figure 10.** Effect of viscosity and gas wall losses on SOA concentrations for several experiments. The black line corresponds to SMPS measurements, the red lines correspond to modeled SOA concentrations for the non-viscous aerosol assumption, the blue lines correspond to modeled SOA concentrations for the viscous aerosol assumption, and the green line correspond to modeled SOA concentrations for the viscous aerosol assumption with a decrease by 20 % of volatilities. Solid lines corresponds to simulations assuming no gas wall losses, dotted lines with gas wall losses.