# Peer review of "Simulating secondary organic aerosol from anthropogenic and biogenic precursors: comparison to outdoor chamber experiments, effect of oligomerization on SOA formation and reactive uptake of aldehydes"

_Atmospheric Chemistry and Physics, 2017_

## Referee Comment (RC1) · Anonymous Referee #1 · 26 Mar 2018

The Couvidat et al. manuscript reports on a series of parameterizations implemented in a model for secondary organic aerosol (SOA). Model simulation results are compared with experimental data obtained from the Euphore chamber, using both anthropogenic and biogenic precursors. The model used is the SOAP model, which uses surrogate compounds for each precursor that best reproduce the bulk properties of the SOA formed (e.g., O/C and H/C ratios). SOAP model parameterizations are developed for anthropogenic and biogenic precursors, and to represent oligomerization and acid-

catalyzed uptake of aldehydes. Sensitivity of model simulations to increased particle viscosity and vapor wall loss are considered. While there are some complex processes that are well represented in the model, and these complex processes are treated in a single model framework, the manuscript lacks clear direction. The objectives of the modeling are not clear, nor are the scientific contributions. Development of SOA models has been rapidly advancing over the last 20 years; parameterizations for all of the processes described in this manuscript have been developed and applied. The application of different parameterizations is not particularly novel, nor does it clearly advance the state of the science. It is suggested that this manuscript undergo major revisions prior to publication. The manuscript may be improved by focusing on one of the parameterizations (e.g., oligomerization) and comparing more rigorously to other model parameterizations and a broader suite of published studies. The manuscript should also be read carefully for clarity and grammatical errors. This will also improve the manuscript and increase its potential impact.

Comments: The mechanism parameters for aromatics were developed largely from chamber studies that are now 10+ years old. For at least some of the compounds of interest, more recent data are available. For example: Hildebrandt et al., ACP 2015 (toluene) and Li et al., ACP 2016 (aromatics, low NOx). The same is true of the parameters used to represent oligomerization. See for example, Kundu et al., ACP 2016 (oligomers from limonene). There may be good reasons for using the particular studies/data chosen, but those reasons should be articulated.

Abstract, lines 6-7: replace "were" with "where"

p.3, line 20: It is recommended that the authors be more specific about the ranges of conditions covered by the experiments, particularly for atmospherically-relevant conditions that are outside the available datasets.

p.5, line 3-4: Chamber data studies are referenced twice.

p. 7, lines 27-30: How are "short", "big", and "bigger" oligomers defined?

p. 11, lines 11-15: Can more be said about the underlying reasons for differences between the model simulations in this work with those of Santiago et al. 2012? Particularly from the perspective of describing the likely processes in the experimental studies.

p. 11, lines 31-33: It is suggested that this section be rewritten to clarify that the model simulations are over/under predicting in different chemical regimes. It is not clear as written. p. 15: The discussion of the consideration of vapor wall loss is incomplete and perhaps misleading. The measurement/model agreement will reflect both the model parameterizations (as indicated), but also the experimental conditions. In this case, underestimation by inclusion of vapor wall loss may largely be due to the fact that vapor wall loss was negligible in the Euphore chamber under the experimental conditions. There is not good scientific support for broadly applying a 3-fold decrease in SVOC volatility and a single vapor wall loss parameterization for all SOA models developed using the Odum approach as applied to all Teflon chamber studies. The extent of wall loss involves competing kinetic processes, and will be highly dependent on the chamber and the experimental conditions.

---

## Referee Comment (RC2) · Anonymous Referee #2 · 29 Mar 2018

Couvidat et al. develop updates to a secondary organic aerosol (SOA) model and report their findings on how those updates perform against chamber SOA measurements made with mixtures of biogenic and anthropogenic precursors. They find that, in general, the updates help improve the model-measurement comparison but offer nuanced insights on the role of NO, oligomerization, vapor wall losses, reactive uptake of pinonaldehyde, and particle phase on SOA formation.

[Figure]

The area of study undertaken by the authors is very important, that of understanding the processes that determine the formation, composition, evolution, and properties of SOA from oxidation of organic precursors. Findings here will help develop simplified mechanisms for atmospheric models. However, the manuscript in its current form does not do well in communicating the methods and, in some cases, the results and implications of the modeling efforts (see some comments below for details). What makes the manuscript even harder to understand is that there are numerous grammatical mistakes and phrasing/style issues. These need to be fixed before the manuscript can be reviewed again, in addition to achieving the quality desired in a journal like Atmospheric Chemistry and Physics. Although I believe this is important and novel work, I cannot make a judgement at this point based on the manuscript submitted for review. Hence, I do not recommend publication in ACP until the issues I discuss below are resolved, the presentation quality is significantly improved, and the manuscript is sent out for review again.

Major Comments:

1. Details on experimental methods and data: There is very little description of the experimental methods used to provide context to the modeling in this work. For example, how big is the Euphore facility? Given the size, was it correct to use the same wall loss rate as that used by the Caltech chamber to model vapor wall losses? Was it a Teflon chamber? What was the motivation to use a mix of precursors instead of using a single precursor? Were these photooxidation experiments or ozonolysis experiments (especially for the biogenic mixture)? Was an OH precursor used and if yes, which one? What photochemical ages were achieved? Was ozone produced? If yes, how much? Were these experiments seeded? What instrumentation was used to measure aerosol mass concentrations? How were the data corrected for wall losses? What are the uncertainties in the measurement data? Answers to these questions and more that bear relevance to the modeling need to be provided as part of section 2.1.

2. Details on modeling methods: Various details of the modeling approach are missing

that make it hard to understand the simulated processes. For example, (page 4, lines 30-32), why was the wall loss rate for vapors used in this work based on the Caltech chamber. A vapor wall loss rate could be estimated for the Euphore facility based on the calculations laid out in the supporting information of Zhang et al. (PNAS, 2014). Why was this not done? Also, the vapor wall loss rate only defines the loss rate of vapors. The affinity of these compounds to stick to the walls was modeled by Zhang et al. (PNAS, 2014) and later shown by Krechmer et al. (ES&T, 2016) to be a function of the vapor pressure of the species. Was this modeled similarly? In the mechanism section (page 5, lines 1-31), the abbreviations for the different species are obvious but it would be worthwhile to explicitly specify them, e.g., API is never defined. Perhaps, include this information in the tables as a legend. Why is only the number mean used to model particle size and not use the entire aerosol size distribution?

3. Structure, grammar, phrasing, and style: In terms of structure, I did not understand the order of the figures. I would recommend that the figure numbers be ordered in the order they show up in the manuscript. Further, the grammar, phrasing and style could be significantly improved. Here is a sample of mistakes I found just in Section 3.3 on page 13: (a) line 8: 'they dynamic of the uptake', (b) line 18: 'specific of the compound', and (c) line 18-19: 'probably provides a good estimate and order of magnitude.'. The manuscript is littered with such mistakes.

Minor comments: 1. Page 2, line 13: Consider citing the chemical transport model study of Cappa et al. (ACP, 2016) that simulated the influence of vapor wall losses on organic aerosol (OA) mass concentrations in urban areas.

2. Page 2, line 14: Jathar et al. (ACP, 2016) have showed – similar to the findings in this work – that oligomerization may not play an important role in affecting SOA mass concentrations but may change the SOA composition. Consider citing.

3. Page 3, line 17: How big is the Euphore facility?

4. Page 4, line 29: What does chamber is closed mean?

5. Page 5, line 23-29: Are the different O:Cs possibly from differences in OA mass concentrations in the different experiments?

6. Page 11, line 7-8: Is particle number or mass used to determine particle wall loss rates? Why are the particle wall loss rates different for with and without oligomerization?

7. Page 11, line 28: Could the differences in SOA formation be explained as a function of the VOC/NOx ratio expressed in ppbC/ppb, similar to previous work?

8. Page 12, line 17: Use 'fragmentation' instead of 'fractionalization'.

9. Page 14, line 7-8: Sentence is unclear and needs more explanation.

10. Page 14, line 14-15: What figure shows a factor of 3 difference for the simulation that includes the loss of vapors to the walls.

11. Page 14, line 23: What do stoichiometric coefficients mean here?

---

## Author Comment (AC1) · 5 Jun 2018

First of all, we would like to thank the reviewers for their comments. The response to the comments are showed after each comment (in italics).

*Anonymous Referee #1*

*The Couvidat et al. manuscript reports on a series of parameterizations implemented in a model for secondary organic aerosol (SOA). Model simulation results are compared with experimental data obtained from the Euphore chamber, using both anthropogenic and biogenic precursors. The model used is the SOAP model, which uses surrogate compounds for each precursor that best reproduce the bulk properties of the SOA formed (e.g., O/C and H/C ratios). SOAP model parameterizations are developed for anthropogenic and biogenic precursors, and to represent oligomerization and acid catalyzed uptake of aldehydes. Sensitivity of model simulations to increased particle viscosity and vapor wall loss are considered. While there are some complex processes that are well represented in the model, and these complex processes are treated in a single model framework, the manuscript lacks clear direction. The objectives of the modeling are not clear, nor are the scientific contributions. Development of SOA models has been rapidly advancing over the last 20 years; parameterizations for all of the processes described in this manuscript have been developed and applied. The application of different parameterizations is not particularly novel, nor does it clearly advance the state of the science. It is suggested that this manuscript undergo major revisions prior to publication. The manuscript may be improved by focusing on one of the parameterizations (e.g., oligomerization) and comparing more rigorously to other model parameterizations and a broader suite of published studies. The manuscript should also be read carefully for clarity and grammatical errors. This will also improve the manuscript and increase its potential impact.*

We agree that the introduction and the goals need to be clearer. The introduction was rewritten to emphasize the fact that this study is part of an update of the H2O mechanism using the molecular surrogate approach and that the paper aim at developing (and also evaluating) a SOA mechanism including various processes which are often not taken into account together. We think that only taking into account oligomerization would significantly decrease the interest of the study (and would not have been possible without reformulating the H2O mechanism). Moreover, our knowledge oligomerization has been scarcely in 3D air quality models as well as the impact of viscosity or the dynamic uptake of aldehyde onto acidic aerosols.

*Comments:*

*The mechanism parameters for aromatics were developed largely from chamber studies that are now 10+ years old. For at least some of the compounds of interest, more recent data are available. For example: Hildebrandt et al., ACP 2015 (toluene) and Li et al., ACP 2016 (aromatics, low NOx). The same is true of the parameters used to represent oligomerization. See for example, Kundu et al., ACP 2016 (oligomers from limonene). There may be good reasons for using the particular studies/data chosen, but those reasons should be articulated.*

This work was initiated a few years ago when the article of Li et al. 2016 was not published and the algorithms are fitted on published Odum's curves data (Hildebrandt et al. do not report any Odum

Curve). Although some of the data used came from old chamber studies, we think that those data are not outdated. As for oligomerization, some parameterizations for 3D modeling were already based on the same experiments of kalberer. A comparison of the results between these parameterizations is even done in the study. Moreover, no quantitative data are present in Kundu et al. 2012 on the evolution of molar masses with time (that were needed for the development of the model).

*Abstract, lines 6-7: replace "were" with "where"*

Corrected

*p.3, line 20: It is recommended that the authors be more specific about the ranges of conditions covered by the experiments, particularly for atmospherically-relevant conditions that are outside the available datasets.*

The range of conditions covered by each of the experiments were summarized in Tables 1 and 2. A sentence is added to refer to these tables, in case this was not enough visible in the text.

*p.5, line 3-4: Chamber data studies are referenced twice.*

The sentence was reformulated.

*p. 7, lines 27-30: How are "short", "big", and "bigger" oligomers defined?*

Definitions are added into the text.

Short oligomers: oligomers of 2 to 4 monomers blocks that can be formed quickly during the first hours

Big oligomers: more than 4 blocks of monomers

Bigger oligomers: oligomers with higher molar masses

*p. 11, lines 11-15: Can more be said about the underlying reasons for differences between the model simulations in this work with those of Santiago et al. 2012? Particularly from the perspective of describing the likely processes in the experimental studies.*

There could be many reasons for the differences between the two studies as the box models used are very different: dynamic approach vs the equilibrium approach, comparison to uncorrected with simulation of depositions to wall vs comparison to corrected values without simulating deposition and how the mechanism in itself was developed or how CMAQ treat the partitioning of secondary organic aerosol. A comparative study would be needed to compare the code of the two models and the data used.

*p. 11, lines 31-33: It is suggested that this section be rewritten to clarify that the model simulations are over/under predicting in different chemical regimes. It is not clear as written.*

The paragraph was rewritten to improve clarity.

*p. 15: The discussion of the consideration of vapor wall loss is incomplete and perhaps misleading. The measurement/model agreement will reflect both the model parameterizations (as indicated), but also the experimental conditions. In this case, underestimation by inclusion of vapor wall loss may largely be due to the fact that vapor wall loss was negligible in the Euphore chamber under the experimental conditions. There is not good scientific support for broadly applying a 3-fold decrease in SVOC volatility and a single vapor wall loss parameterization for all SOA models developed using the*

*Odum approach as applied to all Teflon chamber studies. The extent of wall loss involves competing kinetic processes, and will be highly dependent on the chamber and the experimental conditions*

Due to the uncertainty on the parameter, we initially intended to simply investigate what could be the impact of vapor losses and how it could impact SOA. A value for the vapor loss rate for Euphore is now estimated. Moreover, a mistake producing too much wall deposition was also found and corrected. Section 3.4 as changed.

*Anonymous Referee #2

*Couvidat et al. develop updates to a secondary organic aerosol (SOA) model and report their findings on how those updates perform against chamber SOA measurements made with mixtures of biogenic and anthropogenic precursors. They find that, in general, the updates help improve the model-measurement comparison but offer nuanced insights on the role of NO, oligomerization, vapor wall losses, reactive uptake of pinonaldehyde, and particle phase on SOA formation.*

*The area of study undertaken by the authors is very important, that of understanding the processes that determine the formation, composition, evolution, and properties of SOA from oxidation of organic precursors. Findings here will help develop simplified mechanisms for atmospheric models. However, the manuscript in its current form does not do well in communicating the methods and, in some cases, the results and implications of the modeling efforts (see some comments below for details). What makes the manuscript even harder to understand is that there are numerous grammatical mistakes and phrasing/style issues. These need to be fixed before the manuscript can be reviewed again, in addition to achieving the quality desired in a journal like Atmospheric Chemistry and Physics. Although I believe this is important and novel work, I cannot make a judgement at this point based on the manuscript submitted for review. Hence, I do not recommend publication in ACP until the issues I discuss below are resolved, the presentation quality is significantly improved, and the manuscript is sent out for review again.*

*Major Comments:*

*1. Details on experimental methods and data: There is very little description of the experimental methods used to provide context to the modeling in this work. For example, how big is the Euphore facility? Given the size, was it correct to use the same wall loss rate as that used by the Caltech chamber to model vapor wall losses? Was it a Teflon chamber? What was the motivation to use a mix of precursors instead of using a single precursor? Were these photooxidation experiments or ozonolysis experiments (especially for the biogenic mixture)? Was an OH precursor used and if yes, which one? What photochemical ages were achieved? Was ozone produced? If yes, how much? Were these experiments seeded? What instrumentation was used to measure aerosol mass concentrations? How were the data corrected for wall losses? What are the uncertainties in the measurement data? Answers to these questions and more that bear relevance to the modeling need to be provided as part of section 2.1.*

Yes, it's true that more information would help to introduce this study. The section 2.1 has been rewritten to provide more information about all these questions.

*2. Details on modeling methods: Various details of the modeling approach are missing that make it hard to understand the simulated processes. For example, (page 4, lines 30-32), why was the wall loss*

*rate for vapors used in this work based on the Caltech chamber. A vapor wall loss rate could be estimated for the Euphore facility based on the calculations laid out in the supporting information of Zhang et al. (PNAS, 2014). Why was this not done? Also, the vapor wall loss rate only defines the loss rate of vapors. The affinity of these compounds to stick to the walls was modeled by Zhang et al. (PNAS, 2014) and later shown by Krechmer et al. (ES&T, 2016) to be a function of the vapor pressure of the species. Was this modeled similarly? In the mechanism section (page 5, lines 1-31), the abbreviations for the different species are obvious but it would be worthwhile to explicitly specify them, e.g., API is never defined. Perhaps, include this information in the tables as a legend. Why is only the number mean used to model particle size and not use the entire aerosol size distribution?*

We added more information on the computation of vapor wall losses in section 3.4. Vapor wall losses were calculated similarly to Krechmer et al. and Zhang et al.

Due to the uncertainty on the parameter, we initially intended to simply investigate what could be the impact of vapor losses and how it could impact SOA. A value for the vapor loss rate for Euphore is now estimated. Moreover, a mistake producing too much wall deposition was also found and corrected. Section 3.4 has changed.

The abbreviations are now defined in the text and in the table for the mechanisms.

The following sentence was also added into the text:

"As modeling properly nucleation and coagulation of particles would be needed to simulate adequately the size distribution of particles, particles were gathered inside a single diameter bin."

*3. Structure, grammar, phrasing, and style: In terms of structure, I did not understand the order of the figures. I would recommend that the figure numbers be ordered in the order they show up in the manuscript. Further, the grammar, phrasing and style could be significantly improved. Here is a sample of mistakes I found just in Section 3.3 on page 13: (a) line 8: 'they dynamic of the uptake', (b) line 18: 'specific of the compound', and (c) line 18-19: 'probably provides a good estimate and order of magnitude.'. The manuscript is littered with such mistakes.*

The order of figures was changed, following the reviewer advices.

Several mistakes and grammar errors were corrected. A spell check by the editors will be asked for to track any remaining mistakes.

*Minor comments:*

*1. Page 2, line 13: Consider citing the chemical transport model study of Cappa et al. (ACP, 2016) that simulated the influence of vapor wall losses on organic aerosol (OA) mass concentrations in urban areas.*

Reference added.

*2. Page 2, line 14: Jathar et al. (ACP, 2016) have showed – similar to the findings in this work – that oligomerization may not play an important role in affecting SOA mass concentrations but may change the SOA composition. Consider citing.*

Reference added.

*3. Page 3, line 17: How big is the Euphore facility?*

Details were added in section 2.1. The volume of Euphore is 202 $m^3$.

*4. Page 4, line 29: What does chamber is closed mean?*

"closed" replaced by "enclosed by the retractable steel housing." Explanation are added in section 2.15. *Page 5, line 23-29: Are the different O:Cs possibly from differences in OA mass concentrations in the different experiments?*

The mean O/C may indeed change for different concentrations of OA. However, the precursors and the chemical regimes are probably the most important elements. The range of H/C and O/C from the cited studies are obtained from a variety of conditions with various organic aerosol loading.

*6. Page 11, line 7-8: Is particle number or mass used to determine particle wall loss rates? Why are the particle wall loss rates different for with and without oligomerization?*

In these simulations, the wall deposition rate was constrained to reproduce with the model the decrease of SOA volume concentrations (measured with the SMPS) during the last hours of the experiments. As the computed evolution of SOA concentrations during the last hours can be slightly different with or without oligomerization, the wall deposition rate used with and without oligomerization are different.

*7. Page 11, line 28: Could the differences in SOA formation be explained as a function of the VOC/NOx ratio expressed in ppbC/ppb, similar to previous work?*

The following sentence was added to justify the use of the chemical regime ratio

The chemical regime ratio was used instead of the VOC/NOx because in this study a mixture of VOC (and not a single VOC) was present in the chamber. The chemical regime ratio takes into account the reactivity of the compounds and can therefore be used to compare different experiments with different mixtures of VOC.

*8. Page 12, line 17: Use 'fragmentation' instead of 'fractionalization'.*

Corrected

*9. Page 14, line 7-8: Sentence is unclear and needs more explanation.*

"For non-viscous aerosols, deposition of particles to wall can lead to the evaporation of SVOC due to a decrease of the absorbing mass." Replaced by "For non-viscous, deposition of particles to the walls lead to a decrease of the absorbing mass (mass of the organic aerosol). As the gas/particle partitioning is proportional to the absorbing mass, SVOC will evaporate to maintain the gas/particle partitioning whereas this evaporation will be limited for a viscous aerosol."

*10. Page 14, line 14-15: What figure shows a factor of 3 difference for the simulation that includes the loss of vapors to the walls.*

Reference to the figure added.

*11. Page 14, line 23: What do stoichiometric coefficients mean here?*

Detail added.

---

## Editor Decision (ED1)

Your following responses to the referee's comments are not clear. They do not reflect any changes.

"satisfactory results (bias lower than 20\%)" replaced by "satisfactory results (bias lower than 20%)"

"Oligomerization was found to have a strong effect on SOA composition" changed to "Oligomerization was found to have a strong effect on SOA composition"

**Editor**

General comments:

1) As pointed out by both reviewers before, the paper lacks a clear statement why the current parameterization is better than previous ones. At many places, the fits and estimates seem arbitrary and empirical. The comparison to previous model results using other parameterizations is often very descriptive without interpretations. In order to clarify the message of the manuscript, I suggest clarifying throughout the paper the novelty and improvement based on scientific explanations, i.e. going beyond 'our model results are closer to observations' – which might be a coincidence.

2) I agree with Reviewer #2 that the manuscript is often hard to follow. I think some of this is due to an often sloppy reference to your various parameterizations. For example, 'the extended parameterization' is nowhere clearly defined, and instead of writing the 'HSO4-' approach, it should be referred to the corresponding equation etc. I suggest going carefully through the manuscript and to be as clear as possible in such cross references between sections.

**Specific comments**

p. 1, l. 20: then → than

p. 4, l. 21: "all the range of values" – better: "The full range of values"

p. 4, l. 24: reach → reached

p. 4, l. 28: 'humidities' misspelled

p. 5, l. 7: either 'leads' or 'led'

p. 6, l. 0: "A new mechanism was developed for SOA formation from toluene (TOL), o-xylene (XYL) and trimethylbenzene (TMB)"

Isn't that statement a bit pretentious? Did you reformulate all oxidation steps etc in the oxidation of these precursors semi-/non-volatile compounds? Or did you just update the yields and stoichiometric coefficients using a previous approach, e.g. Odum or the VBS framework?

p. 7, l. 16: these results

p. 7, l. 23: "Oligomers are represented by simple species to know if "monomer blocks" are present mainly as oligomers or as monomers."

I don't understand this sentence.

p. 7, l. 28: "the same compound AS.."

p. 9, l. 6: "To do that": What does it refer to? 'To measure that' or 'to estimate that'? – It might help to clarify in l. 5 "in their study" (not to be confused with "in this present study")

p. 9, l. 10: Here is the first place where you introduce the term 'extended parameterization'. Some explanation would help why it is extended and called like that.

p. 9, l. 28: ..did not find…

p. 9, l. 29: "…with a dimer formation that cannot react further…" – better: …formation of dimers that cannot react further .." (a 'formation' cannot react further)

p. 9, l. 34: "that for those 4 dimers are not formed from particle-phase reaction" – remove 'for'

p. 10, l. 3-4: "In this study, the second order parameterization was used for simulations. In case of the oligomerization inside an aqueous
acidic phase, a kinetic rate of 8.76 a[H+] should be used to take into account the effect of acidity on oligomerization."

a) What is the second order parameterization? Cn you refer to a previous equation or paragraph?
b) The connection of the two sentences is not clear.

p. 10, l. 24: Liggio and Li (2006b) did not evaluate

p. 11, l. 5: Why is it called Jtrans here?

p. 11, l. 9: What assumption can be/are made for the back reaction of the reversible process?

p. 11, l. 21-27, and Table 7: It is not clear which if the references in the text were used for the data in Table 7. At the very least add references to the values in Table 7.

p.12, l. 31: "…the wall deposition rate […] are different…"   - either 'rate is different' or 'rates are different'

p. 13, l. 2: "…to ensure that effect of changes on deposition remain low."

Clarify this statement. How low is 'low' and why would it make sense to assume different deposition losses with and without oligomerization in general? Wouldn't it be more useful to use the same (averaged?) deposition loss rate for both scenarios?

p. 13, l. 3-5: "For the biogenic experiments, the model gives good results (bias lower than 20%) with or without oligomerization for all experiments with slightly better results without oligomerization for experiment B5 and slightly better results with oligomerization for experiment B1."

Can we learn something from this finding? What is reasoning behind it?

p. 14, l. 15: "Assuming aging leads to a slight decrease of SOA mass due to fragmentation for toluene SOA or an increase of concentrations due to functionalization for TMB and _-pinene SOA."

This sentence seems incomplete.

p. 14, l. 18-27: Can you give any recommendation on which of the parameterization should be used in future modeling studies? And why?

p. 14, l. 31: the partitioning of monomers IS sensitive

p. 15, l. 2: "The pH and HSO4 parameterizations and the parameterization of Pun and Seigneur (2007) assuming equilibrium"

At this point, the reader might not remember what those are. Please refer to the equations.

p. 15, l. 9: same results AS assuming no uptake

p. 15, l. 13: "acidic acid of pinonaldehyde" -- ?

p. 17, l. 18: Redefine here briefly the term 'chemical regime ratio'

p. 18, l. 1: 'decrease' of what?

p. 18, l.3: ans → and

p.18, l. 4: "This decrease could however be compensated by decreasing the volatility 5 of SVOC by 20%."

It is not clear what you want to say here.

---

## Author Response (AR2)

*Referee #2:*

*The authors have made significant improvements to the manuscript since their first submission and it is increasingly clear what the contribution of this work is to the literature. However, there are still a lot of presentation issues - particularly with the methods sections - with the revised manuscript that make it hard to evaluate the results from the work. For instance, there is little detail for each process that is modeled in terms of what has been done before, what is new, and what the different terminology and equations stand for. Even within each process, the structure of the mathematical formulation is not very clear and at times hard to follow. Below, I list a few examples:*

To our knowledge, the main studies on "what has been done before" are cited in the introduction and are described in the method section. As this study cover several subjects that has been scarcely taken into account. As stated into the text, all the parameterizations described in the text are new (and explanations were given on how they differ). The results given by the parameterizations for oligomerizations and for the uptake of pinonaldehyde were compared to the results obtained with existing parameterizations.

However, the text was modified to take into account each comment.

*1. See paragraph 3 in the abstract. The authors make qualitative judgements about their model results, e.g., 'satisfactory results', 'strong effect', and 'too significant', without being specific or quantitative. The last sentence in paragraph 3 in the abstract about 'less volatile or more reactive aldehydes...' is hard to connect to the earlier sentence.*

"satisfactory results (bias lower than 20\%)" replaced by "satisfactory results (bias lower than 20%)

 "The uptake of pinonaldehyde (which is15 a high volatility SVOC) onto acidic aerosol was found to be too to be significant under atmospheric conditions" changed to "The uptake of pinonaldehyde (which is a high volatility SVOC) onto acidic aerosol was found to be too slow to be significant under atmospheric conditions (no significant amount of SOA formed after 3 days of evolution)"

"Oligomerization was found to have a strong effect on SOA composition" changed to "Oligomerization was found to have a strong effect on SOA composition"

"Less volatile or more reactive aldehydes could nevertheless react with acidic aerosols." Replaced by "he uptake of aldehydes could nevertheless be an important SOA formation pathway for less volatile or more reactive aldehydes then pinonaldehyde."

*2. Section 2.2.1: What xylene are the authors referring to? m-, o-, or p-xylene? Why is table 7 referred to after table 3? Shouldn't it be table 4?*

It is already written in section 2.2.1 and the abstract that xylenes correspond o-xylene in this study. Several references to o-xylenes are added to the text. The order of tables is corrected.

*3. Page 7, line 32: What is the meaning of the sentence 'chemical rates are more consistent with*

*thermodynamic equilibrium by computing rates using activities'? What is activity and how is it defined here?*

The text was modify to provide more information:

"In this study, the net flux of oligomerization $J_{oligo}$ is computed using activities. Activity is often seen as the "apparent concentration" of a compound in thermodynamics. It is linked to the chemical potential (molar Gibbs free energy of a particular component) by the following equation:

$$a_i = exp(\frac{\mu_i - \mu_i^0}{RT})$$

with $a_i$ the activity of compound i, $\mu_i$ is the chemical potential of compound i and $\mu_i^0$ the chemical potential under standard conditions, R is the gas constant, T is thermodynamic temperature.

Activities (calculated here on the mole fraction basis) are used instead of concentrations for two main reasons. First, chemical rates are more consistent with thermodynamic equilibrium by computing rates using activities. For example, in the case of a simple one product (A) giving one product (B) equilibrium reaction, if chemical reactions are written using concentrations, the net flux of reaction J would be computed with the following equation:

$$J = k_1 C_A - k_{-1} C_B$$

with $k_1$ the forward kinetic parameter, $k_{-1}$ the reverse kinetic parameter, $C_A$ the concentration of compound A and $C_B$ the concentration of compound B. At equilibrium, J would be equal to zero and the equilibrium constant would then correspond to the ratio of concentrations instead of a ratio of activities.

This paradox can be lifted by using activities instead of concentrations. Second, some studies (Madon and Iglesia, 2000; Rahimpour, 2004) expressed the need to compute chemical rates using activities and showed that better results are obtained for non-ideal systems.

The net flux of oligomerization $J_{oligo}$ is therefore computed with the following equations:

$$J_{oligo} = -\frac{dX_{a,monomer}}{dt} = k_{oligo} a_{a,monomer} - k_{reverse} a_{a,oligomer}$$

with $X_{a,monomer}$ the molar fraction of compound a, $a_{a,monomer}$ the activity on a molar fraction basis of compound a and $a_{a,oligomer}$ the activity on a molar fraction basis of the oligomer formed from compound a. Activities are computed with the AIOMFAC model (Zuend et al., 2008, 2011; Zuend and Seinfeld, 2012; Ganbavale et al., 2015)."

*4. Page 8: What are the different terms in equation 5? What do the '-1' and '-2' superscripts mean?*

The terms were defined previously in the text (except for $a_{H2O}$ for which the definition is added). -1 and -2 does not represent superscripts but "minus 1" and "minus 2" in power functions.

The equations were modified to improve the readability. For example:

$$(K_{oligo}^{eq})^{m_{oligo}-1} = \frac{a_{a,oligomer}(a_{H_2O})^{m_{oligo}-1}}{a_{a,monomer}(a_{monomer})^{m_{oligo}-1}}$$

*5. There were a lot of grammatically incorrect phrases, which made it hard to understand their meaning, e.g., 'uptake kinetic rate of aldehydes', 'particle but a kinetic of uptake', 'kinetic of uptake can be linked to the kinetic of transformation'.*

Corrected.

*6. What does the product BiA0D represent? It is introduced on page 10 without any context.*

BiA0D = pinonaldehyde (notation from the H$^2$O mechanism). The text was modified to better explain this.

*7. What does 'according to experimental conditions' on page 10, line 11 mean?*

"the pH of particles and activities of compounds were computed with AIOMFAC (…) according to experimental conditions." Changed to "the pH of particles and activities of compounds were computed with AIOMFAC (…) depending on the conditions (humidity, temperature, concentrations, etc…) of the experiments of Liggio and Li (2006b)."

---

## Author Response (AR3)

*Your following responses to the referee's comments are not clear. They do not reflect any changes.*
*"satisfactory results (bias lower than 20\%)" replaced by "satisfactory results (bias lower than 20%)"*
*"Oligomerization was found to have a strong effect on SOA composition" changed to*
*"Oligomerization was found to have a strong effect on SOA composition (oligomers were estimated to account for up to 78% of the SOA mass)"*

Sorry for the quick copy/paste.

The changes were :

- "satisfactory results" replaced by "satisfactory results (bias lower than 20%)"

- "Oligomerization was found to have a strong effect on SOA composition" changed to "Oligomerization was found to have a strong effect on SOA composition (oligomers were estimated to account for up to 78% of the SOA mass)""

*Editor General comments:*

*1) As pointed out by both reviewers before, the paper lacks a clear statement why the current parameterization is better than previous ones. At many places, the fits and estimates seem arbitrary and empirical. The comparison to previous model results using other parameterizations is often very descriptive without interpretations. In order to clarify the message of the manuscript, I suggest clarifying throughout the paper the novelty and improvement based on scientific explanations, i.e. going beyond 'our model results are closer to observations' – which might be a coincidence.*

The main of this study was to introduce a more realistic representation of SOA phenomena more than bringing the results closer to observations. The section 3.1 is voluntary descriptive as this part concerns the evaluation of the mechanism to ensure the good performances of the mechanism for conditions different from the conditions under which it was developed. This objective was repeated at the beginning of section 3.1.

For Oligomerization and the uptake of pinonaldehyde, the differences with previous parameterizations (and the reasons for which we expect the parameterizations to be more realistic) were explained at the beginning of the section 2.2.2 and 2.2.3. We modified the text to repeat this information in the conclusion as well as the introduction to clarify the novelty of such parameterizations.

Several details were added into the text.

*2) I agree with Reviewer #2 that the manuscript is often hard to follow. I think some of this is due to an often sloppy reference to your various parameterizations. For example, 'the extended parameterization' is nowhere clearly defined, and instead of writing the 'HSO4-' approach, it should be referred to the corresponding equation etc. I suggest going carefully through the manuscript and to be as clear as possible in such cross references between sections.*

The oligomerization section was rewritten to better describe the parameterizations and references to equations were added.

Specific comments

*p. 1, l. 20: then → than* Corrected

*p. 4, l. 21: "all the range of values" – better: "The full range of values"* Corrected

*p. 4, l. 24: reach → reached* Corrected

*p. 4, l. 28: 'humidities' misspelled* Corrected

*p. 5, l. 7: either 'leads' or 'led'* Corrected

*p. 6, l. 0: "A new mechanism was developed for SOA formation from toluene (TOL), o-xylene (XYL) and trimethylbenzene (TMB)" Isn't that statement a bit pretentious? Did you reformulate all oxidation steps etc in the oxidation of these precursors semi-/non-volatile compounds? Or did you just update the yields and stoichiometric coefficients using a previous approach, e.g. Odum or the VBS framework?*

It is really a new mechanism (new formulation of the mechanism) and not simply an update of some parameters.

*p. 7, l. 16: these results* Corrected

*p. 7, l. 23: "Oligomers are represented by simple species to know if "monomer blocks" are present mainly as oligomers or as monomers." I don't understand this sentence.*

This section was rewritten to improve clarity.

*p. 7, l. 28: "the same compound AS.."* Corrected

*p. 9, l. 6: "To do that": What does it refer to? 'To measure that' or 'to estimate that'? – It might help to clarify in l. 5 "in their study" (not to be confused with "in this present study")*

"To do that" replaced by "to represent oligomerization"

"in this study" replaced by "in their study"

*p. 9, l. 10: Here is the first place where you introduce the term 'extended parameterization'. Some explanation would help why it is extended and called like that.*

The section was rewritten to better introduce this parameterization

*p. 9, l. 28: ..did not find…*

Corrected

*p. 9, l. 29: "…with a dimer formation that cannot react further…" – better: …formation of dimers that cannot react further .." (a 'formation' cannot react further)*

Corrected

*p. 9, l. 34: "that for those 4 dimers are not formed from particle-phase reaction" – remove 'for'*

Corrected

*p. 10, l. 3-4: "In this study, the second order parameterization was used for simulations. In case of the oligomerization inside an aqueous acidic phase, a kinetic rate of 8.76 a[H+] should be used to take into account the effect of acidity on oligomerization."*

*a) What is the second order parameterization? Cn you refer to a previous equation or paragraph?*

Referrence to the reduced parameterization was added

*b) The connection of the two sentences is not clear.*

The second sentence was moved to connect with the previous discussion on the effect of pH on Oligomerization.

*p. 10, l. 24: Liggio and Li (2006b) did not evaluate*

Corrected

*p. 11, l. 5: Why is it called Jtrans here?*

Jtrans replaced by ktrans

*p. 11, l. 9: What assumption can be/are made for the back reaction of the reversible process?*

The following sentence is added:

"The reaction is assumed to be complete as according to the parameterization of Pun and Seigneur (2007), the uptake of pinonaldehyde onto an acidic aerosol should be to be an irreversible process."

*p. 11, l. 21-27, and Table 7: It is not clear which if the references in the text were used for the data in Table 7. At the very least add references to the values in Table 7.*

References for SOA yield were added in Table 7.

*p.12, l. 31: "…the wall deposition rate […] are different…" - either 'rate is different' or 'rates are different'*

Corrected

*p. 13, l. 2: "…to ensure that effect of changes on deposition remain low. Clarify this statement. How low is 'low' and why would it make sense to assume different deposition losses with and without oligomerization in general? Wouldn't it be more useful to use the same (averaged?) deposition loss rate for both scenarios?*

Further explanation is provided.

"Usually, in environmental chamber studies, the wall deposition rate is computed based on the evolution of concentrations during the last hours of experiments. However, at the last stage of the experiments, oligomerization could still occur and affect the evolution of concentrations and therefore affect the estimation of the wall deposition rate. The estimated deposition rate could be biased if the evolution of concentrations due to oligomerization at the end of the experiment is significant compared to the deposition rate. In these simulations, the wall deposition rate was constrained to reproduce with the model the decrease of SOA volume concentrations (measured with the SMPS) during the last hours of the experiments. Because the computed evolution of SOA concentrations during the last hours can be slightly different with or without oligomerization, the wall deposition rates used with and without oligomerization are different. To examine the effect of the wall deposition rate, the formation of SOA with oligomerization was also simulated with the deposition rate as computed by the simulation without oligomerization to ensure that effect of changes on deposition remain low (changes of concentrations of a few percent) and that hypothesis on oligomerization does not affect the simulated deposition."

*p. 13, l. 3-5: "For the biogenic experiments, the model gives good results (bias lower than 20%) with or without oligomerization for all experiments with slightly better results without oligomerization for experiment B5 and slightly better results with oligomerization for experiment B1.*

*Can we learn something from this finding? What is reasoning behind it?*

No. We cannot. The differences are below the uncertainties on the input parameter and the measurement of concentrations.

*p. 14, l. 15: "Assuming aging leads to a slight decrease of SOA mass due to fragmentation for toluene SOA or an increase of concentrations due to functionalization for TMB and _-pinene SOA." This sentence seems incomplete.*

The sentence was modified:

"Assuming aging leads to a slight decrease of SOA mass for toluene SOA due to fragmentation while it leads to an increase of concentrations for TMB and α-pinene SOA due to functionalization."

*p. 14, l. 18-27: Can you give any recommendation on which of the parameterization should be used in future modeling studies? And why?*

More details were added in the discussions for oligomerization:

Contrary to Carlton et al. (2010) who represented oligomerization as a first-order irreversible process, oligomerization was represented in the present study (as an effort to provide a more realistic representation) as a second-order reversible reaction unfavored by humid conditions by analogy to oligomerization reactions expected to occur in the particles, such as esterification, hemiacetalization, aldolization, peroxyhemiacetalization. With this new parameterization, oligomerization was shown to have little impact on SOA mass during the experiments. However, oligomerization was found to affect SOA composition as a large part of SOA is constituted (more than 50% for the biogenic experiments) by oligomers according to the simulations. These results seem to be consistent with the high amount of oligomers reported by Gao et al. (2004a), whereas a low amount of oligomers was simulated with the parameterization of Carlton et al. (2010) (below 15% at the end of the experiments). The results of the present study with the new parameterization indicate that oligomerization may be faster than simulated by the parameterization of Carlton et al. (2010) but as the same time may reach an equilibrium. However, more efforts should be deployed to improve this parameterization. Indeed, this parameterization represents a "bulk oligomerization" and does not account for differences in the reactivity of monomers. This parameterization should nonetheless be deployed and tested in 3D air quality models to diagnose the effect of oligomerization on SOA formation.

*p. 14, l. 31: the partitioning of monomers IS sensitive* Corrected

*p. 15, l. 2: "The pH and HSO4 parameterizations and the parameterization of Pun and Seigneur (2007) assuming equilibrium" At this point, the reader might not remember what those are. Please refer to the equations.*

References added.

*p. 15, l. 9: same results AS assuming no uptake*

Corrected.

*p. 15, l. 13: "acidic acid of pinonaldehyde" -- ?*

"acidic acid" replaced by "acidic aerosol"

*p. 17, l. 18: Redefine here briefly the term 'chemical regime ratio'*

Definition added.

*p. 18, l. 1: 'decrease' of what?*

Corrected: "of SOA concentrations"

*p. 18, l.3: ans → and* Corrected

[revised manuscript text omitted]

[a] Oxidants may be present as both reactants and products are the methylperoxy radical and the peroxyacetyl radical respectively.

[Figure]

**Figure 1.** Temporal evolution of molar masses (top) and of the mass fraction of oligomer (down) for the extended (Eq. 1) and reduced (Eq. 3) parameterizations.

[Figure]

**Figure 2.** Kinetic rate of transformation of pinonaldehyde as a function of $m_{H+}$ the molality of ion $H^+$ (left) and $a^{(m)}_{HSO_4^-}$ the activity on a molality basis of ion $HSO_4^-$ (right).

[Figure]

**Figure 3.** Aerosol concentration formation for the biogenic experiments without $SO_2$. Black lines correspond to SMPS measurements.

[Figure]

**Figure 4.** Aerosol concentration formation for the anthropogenic experiments. Black lines correspond to SMPS measurements.

[Figure]

**Figure 5.** Evolution of the SOA yield from $\alpha$-pinene oxidation as a function of time for an organic mass loading of 5 $\mu$g m$^{-3}$. Solid lines correspond to SOA formation with aging. Dashed lines (-.) correspond to SOA formation without aging.

[Figure]

**Figure 6.** Evolution of the SOA yield from toluene oxidation as a function of time for an organic mass loading of 5 $\mu$g m$^{-3}$. Solid lines correspond to SOA formation with aging. Dashed lines correspond to SOA formation without aging.

[Figure]

**Figure 7.** Evolution of the SOA yield from trimethylbenzene oxidation as a function of time for an organic mass loading of 5 $\mu$g m$^{-3}$. Solid lines correspond to SOA formation with aging. Dashed lines correspond to SOA formation without aging.

[Figure]

**Figure 8.** Aerosol concentration formation for the biogenic experiments with SO$_2$. Black lines correspond to SMPS measurements.

[Figure]

**Figure 9.** Yield of formation of SOA, aldehydes and organosulfates with (solid lines) or without (dashed lines) the conversion of aldehydes into organosulfates as a function of time for an organic mass loading of 5 $\mu$g m$^{-3}$ with 2 $\mu$g m$^{-3}$ of sulfates.

[Figure]

**Figure 10.** Effect of viscosity and gas wall losses on SOA concentrations for several experiments. The black line corresponds to SMPS measurements, the red lines correspond to modeled SOA concentrations for the non-viscous aerosol assumption, the blue lines correspond to modeled SOA concentrations for the viscous aerosol assumption, and the green line correspond to modeled SOA concentrations for the viscous aerosol assumption with a decrease by 20 % of volatilities. Solid lines corresponds to simulations assuming no gas wall losses, dotted lines with gas wall losses.